# Noroviruses subvert the core stress granule component G3BP1 to promote viral VPg-dependent translation

Myra Hosmillo[1†], Jia Lu[1†], Michael R McAllaster[2†], James B Eaglesham[1,3], Xinjie Wang[1,4], Edward Emmott[1,5,6], Patricia Domingues[1], Yasmin Chaudhry[1], Tim J Fitzmaurice[1], Matthew KH Tung[1], Marc Dominik Panas[7], Gerald McInerney[7], Nicolas Locker[8], Craig B Wilen[9]*, Ian G Goodfellow[1]*

[1]Division of Virology, Department of Pathology, University of Cambridge, Cambridge, United Kingdom; [2]Department of Pathology and Immunology, Washington University School of Medicine, St. Louis, United States; [3]Department of Microbiology, Harvard Medical School, Boston, United States; [4]Institute for Brain Research and Rehabilitation, South China Normal University, Guangzhou, China; [5]Department of Bioengineering, Northeastern University, Boston, United States; [6]Barnett Institute for Chemical and Biological Analyses, Northeastern University, Boston, United States; [7]Department of Microbiology, Tumor and Cell Biology, Karolinska Institute, Stockholm, Sweden; [8]School of Biosciences and Medicine, University of Surrey, Guildford, United Kingdom; [9]Department of Laboratory Medicine, Yale School of Medicine, New Haven, United States

*For correspondence:
craig.wilen@yale.edu (CBW);
ig299@cam.ac.uk (IGG)

†These authors contributed equally to this work

Competing interests: The authors declare that no competing interests exist.

**Abstract** Knowledge of the host factors required for norovirus replication has been hindered by the challenges associated with culturing human noroviruses. We have combined proteomic analysis of the viral translation and replication complexes with a CRISPR screen, to identify host factors required for norovirus infection. The core stress granule component G3BP1 was identified as a host factor essential for efficient human and murine norovirus infection, demonstrating a conserved function across the *Norovirus* genus. Furthermore, we show that G3BP1 functions in the novel paradigm of viral VPg-dependent translation initiation, contributing to the assembly of translation complexes on the VPg-linked viral positive sense RNA genome by facilitating ribosome recruitment. Our data uncovers a novel function for G3BP1 in the life cycle of positive sense RNA viruses and identifies the first host factor with pan-norovirus pro-viral activity.
DOI: https://doi.org/10.7554/eLife.46681.001

## Introduction

Positive sense RNA viruses rely heavily on host cell factors for all aspects of their life cycle. They replicate on host derived membranous vesicles that are induced following viral infection, the formation of which requires the activity of key membrane bound viral enzymes (*Altan-Bonnet, 2017*). Within the membrane bound viral replication complex, translation of the viral genome and the synthesis of new viral RNA occurs in a highly coordinated process. Positive sense RNA viruses have evolved novel gene expression mechanisms that enable them to overcome the genome size limitations that accompany error-prone replication and which might restrict their overall coding capacity (*Firth and Brierley, 2012*). In addition, viral modification of the host cell translation machinery often provides a competitive advantage allowing for the efficient translation of viral RNA in an environment where competing cellular RNAs are in abundance (*McCormick and Khaperskyy, 2017*). This ability to

**eLife digest** The human norovirus (also known as the winter vomiting bug) is a major cause of gastroenteritis worldwide and is responsible for the deaths of many children in developing countries. Norovirus infections are estimated to have an economic cost of over 60 billion US dollars per year, yet there are no vaccines or drugs to prevent or limit the spread of outbreaks. In most cases norovirus infections last only a few days but, in people with weakened immune systems, infections can last from months to years.

A norovirus particle consists of a molecule of ribonucleic acid (or RNA for short), which contains the genome of the virus, surrounded by a coat of proteins. The virus is unable to multiply on its own and so it infects the cells of its host and hijacks them to make new viral proteins and RNA. Host cells have their own RNA molecules, which provide the instructions that cellular machines called ribosomes need to make proteins. Recent work reported that a norovirus protein called VPg interacts with the host cells' ribosomes to recruit them to produce proteins from the viral genome. However, it remained unclear which host proteins were important for this key stage of the norovirus life cycle.

Hosmillo, Lu, McAllaster et al. combined three different molecular biology and genetic approaches to search for host proteins that help noroviruses to multiply in cells. Any proteins identified in these experiments would be fundamental for the norovirus life cycle, making them potential drug targets for future treatments. The experiments revealed that a protein called G3BP1 was required for noroviruses to multiply efficiently. Previous studies have shown that G3BP1 is a member of a family of proteins that can bind to RNA and play many roles in healthy cells, including helping the cells to adjust the proteins they produce in response to stress. Hosmillo, Lu, McAllister et al. found that G3BP1 helped VPg to recruit ribosomes and the other host components needed to make new proteins from the viral RNA genome.

These findings reveal a new role for G3BP1 in allowing noroviruses to multiply within cells and identifies a potential weakness in the norovirus life cycle. In the future, this work may help researchers to identify new drugs that could control norovirus outbreaks or treat long-term norovirus infections in humans.

DOI: https://doi.org/10.7554/eLife.46681.002

compete with cellular RNAs is particularly important for the initiation of infection where the incoming viral genome may be present at only a single copy per cell.

We have previously described a novel paradigm of viral translation that relies on the interaction of host translation initiation factors with a virus-encoded protein (VPg), covalently linked to the 5' end of the genome of members of the *Caliciviridae* family of positive sense RNA viruses (*Chaudhry et al., 2006*; *Chung et al., 2014*; *Goodfellow et al., 2005*; *Hosmillo et al., 2014*; *Leen et al., 2016*). Unlike the 22-amino acid VPg peptides from picornaviruses, the VPg protein linked to the genomes of caliciviruses is significantly larger and is essential for the translation of viral RNA and viral RNA infectivity (*Goodfellow, 2011*).

Human noroviruses (HuNoV) and sapoviruses (HuSaV) are enteropathogenic members of the *Caliciviridae* family of positive sense RNA viruses, and together cause >20% of all cases of gastroenteritis (GE). They are also a significant cause of morbidity and mortality in the immunocompromised; individuals with genetic immune-deficiencies, cancer patients undergoing treatment and transplant recipients often experience chronic norovirus infections lasting months to years (*van Beek et al., 2017*). The economic impact of HuNoV is estimated to be at least ~\$4.2 billion in direct health care costs, with wider societal costs of ~\$60 billion (*Bartsch et al., 2016*). Despite their socioeconomic impact, we have, until very recently lacked a detailed understanding of much of the norovirus life cycle and many significant questions remain unanswered. HuNoV replicons (*Chang et al., 2006*), a murine norovirus that replicates in cell culture (*Karst et al., 2003*; *Wobus et al., 2004*) and the recent B cell (*Jones et al., 2014*), stem-cell derived organoid (*Ettayebi et al., 2016*) and zebrafish larvae infection models (*Van et al., 2019*), have all provided invaluable tools to dissect the norovirus life cycle. However, due to the technical limitations associated with many of these experimental

systems, in comparison to other positive sense RNA viruses, our knowledge of the intracellular life of noroviruses is significantly lacking (reviewed in *Thorne and Goodfellow, 2014*).

In the current study, we have combined three independent unbiased approaches to identify host factors involved in the norovirus life cycle. Combining experimental systems that incorporated both murine and human noroviruses, allowed the identification of cellular factors for which the function is likely conserved across the *Norovirus* genus. By combining three complimentary approaches, we identify the host protein G3BP1 as a critical host factor required for norovirus VPg-dependent translation, identifying a new role for G3BP1 in virus-specific translation.

## Results

### Comparative analysis of the norovirus translation initiation complex

The MNV and the prototype HuNoV Norwalk virus (NV) VPg proteins are covalently linked to the viral genome via the highly conserved tyrosine residue within an N-terminal DEEYD/E motif found in all calicivirus VPg proteins (*Figure 1A*). In addition, the norovirus VPg proteins contain a highly conserved C-terminal domain which we have shown to be necessary and sufficient for binding to the translation initiation factor eIF4G via an interaction that requires a highly conserved phenylalanine residue (*Figure 1A*) (*Chung et al., 2014*; *Leen et al., 2016*). Using affinity purification on m7-GTP sepharose, we confirmed that the NV VPg protein, as produced during authentic virus replication in a NV replicon bearing cell line, interacts with the cap-binding complex eIF4F (*Figure 1B*). Components of the eIF4F complex, namely the eIF4E cap-binding protein, the eIF4A helicase and the eIF4GI scaffold protein, along with poly-A binding protein (PABP) and eIF3 subunits, were readily purified on m7-GTP sepharose, whereas GAPDH was not. In NV-replicon containing cells, mature VPg was also enriched on m7-GTP sepharose but the NS3 protein, known to have RNA binding and helicase activity (*Li et al., 2018*), was not. Furthermore, we demonstrated that transfection of GFP-tagged versions of either the MNV or NV VPg proteins into 293 T cells allowed for the affinity purification of eIF4F components and that mutations in the eIF4G binding domain of VPg reduced this association (*Figure 1C*).

We next used quantitative mass spectrometry of the affinity purified complexes isolated from cells transfected with the GFP-Tagged VPg proteins to identify host factors specifically enriched on the norovirus VPg protein (*Figure 1D*, *Figure 1—figure supplement 1* and *Supplementary file 1*). Most of the proteins identified were components of the host cell translation complex including ribosomal proteins, translation initiation factors and host RNA binding proteins. These data agrees with but significantly extend our previous observations using a less sensitive multi-step affinity purification approach to characterise host factors associated with the MNV VPg protein only (*Chaudhry et al., 2006*; *Chung et al., 2014*). In addition, we identified hnRNPA1 which we have previously shown to act in norovirus genome circularization (*López-Manríquez et al., 2013*). YBX1, DDX3 and several other proteins that we have previously found to interact with the 5' end of the viral RNA (*Vashist et al., 2012a*) were also enriched on VPg (*Figure 1—figure supplement 1*). To validate a select number of these interactions and to assess whether the interaction of VPg with eIF4G is required for their association with VPg, we performed western blot analysis of complexes purified from cells transfected with either the WT or eIF4G-binding mutants (*Figure 1E*). Except for YBX1, the association of all proteins tested were reduced by the introduction of eIF4G-binding site mutations into the MNV VPg protein. Together, these data extend our previous observations and confirm that the norovirus VPg proteins interact with a complex network of host factors, many of which have been implicated in the host cell translation initiation process.

### Determination of the norovirus replication complex proteome

To further identify the components of the norovirus translation and replication complex, as formed during authentic viral replication in highly permissive cells, we utilised two recombinant infectious MNV strains that carried epitope purification tags within the NS1/2 or NS4 proteins (*McCune et al., 2017*) (*Figure 2A*). The insertion positions were previously identified using a transposon based mutagenesis screen as sites that tolerate insertions, without compromising virus viability (*Thorne et al., 2012*). Our approach was somewhat analogous to that recently published for coronaviruses (*V'kovski et al., 2019*) but instead used stable isotope labelling of permissive cells and the

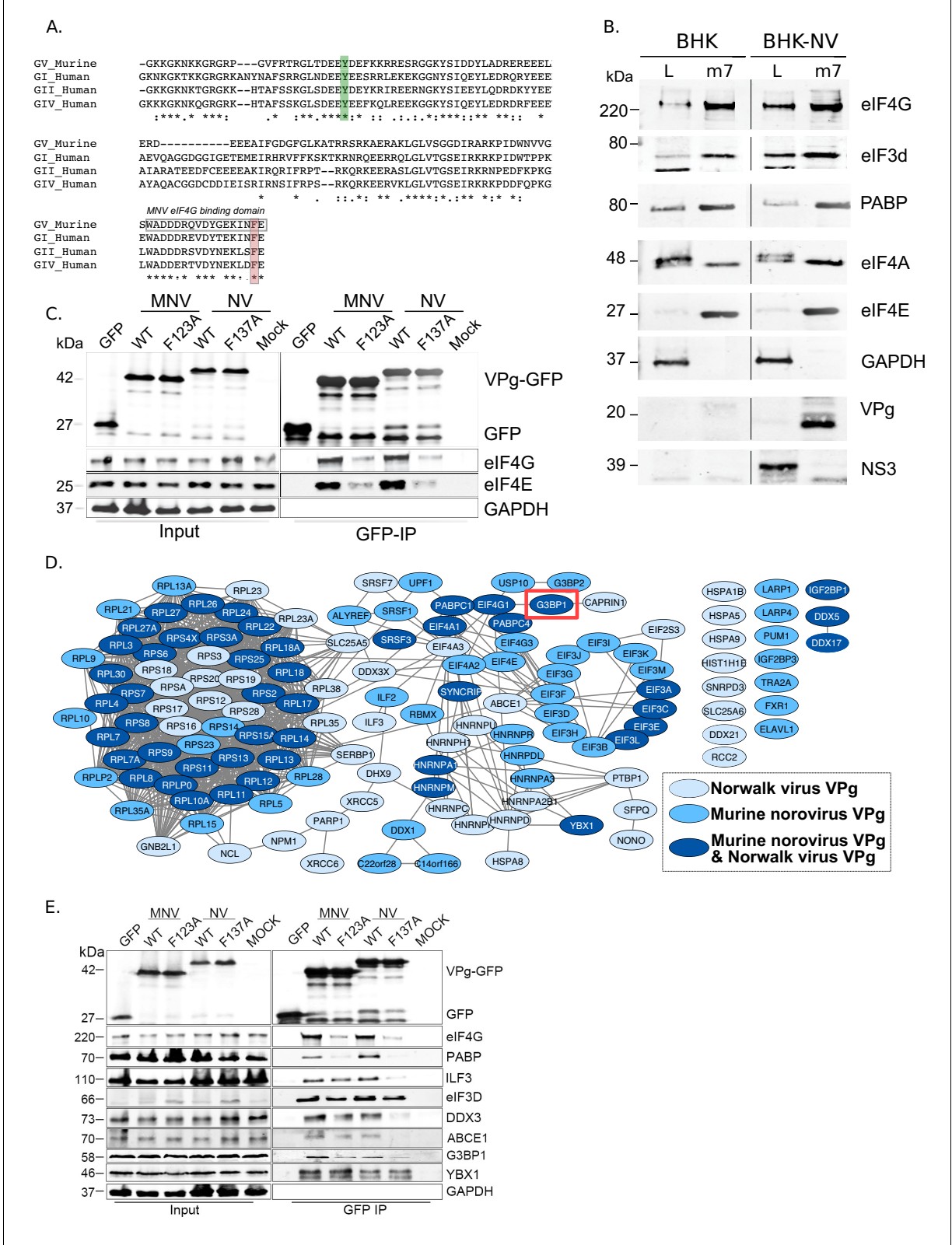

**Figure 1.** The norovirus VPg proteins interacts with ribosome associated translation initiation factors. (**A**) Amino acid sequence alignment of the GV murine norovirus VPg sequences with VPg from representative human noroviruses from GI Norwalk virus (NV), GII, and GIV. The position of the site of RNA linkage to the highly conserved tyrosine residue is highlighted in green. The eIF4G binding motif is boxed and the position of the C-terminal single amino acid change known to interfere with eIF4G binding highlighted in orange. (**B**) m7-GTP sepharose was used to affinity purify eIF4F

*Figure 1 continued on next page*

*Figure 1 continued*

containing complexes from either wild-type BHK cells (BHK) or BHK cells containing the Norwalk virus (NV) replicon (BHK-NV). Samples of the lysate (L) or the affinity purified complexes (m7) were separated by SDS-PAGE then analysed by western blot for the indicate proteins. Molecular mass shown on the left of the gels represent the positions of molecular weight markers. (C) GFP fusion proteins to either the wild type (WT) or C-terminal eIF4G binding domain mutants of the MNV and NV VPg proteins (F123A, F137A) were transfected into human 293 T cells and subjected to immunoaffinity purification using anti-GFP. Samples of the input lysates (Input) and the purified complexes (GFP-IP) were then separated by SDS-PAGE and analysed by western blot analysis for the indicated proteins. Mock transfected cells served as a specificity control. The approximate expected molecular mass of each protein is shown to the left. (D) Quantitative proteomics was used as described in the text to identify host factors that were affinity purified following transfection of GFP-tagged derivative of either the NV or MNV VPg proteins. Proteins specifically enriched in comparison to the GFP control are shown. Data visualisation was performed using Cystoscape (*Shannon et al., 2003*). (E) Western blot analysis of cell extracts (Input) or immunoprecipitated (GFP IP) complexes isolated from cells transfected as described in panel C. For clarity, the molecular masses shown in this panel refer to the expected mass of the protein being examined.

DOI: https://doi.org/10.7554/eLife.46681.003

The following figure supplement is available for figure 1:

**Figure supplement 1.** Host factors binding to the norovirus VPg.

DOI: https://doi.org/10.7554/eLife.46681.004

FLAG affinity purification tag rather than proximity labelling. Unlabelled or stable isotope labelled highly permissive BV2 microglial cells were infected with either wild type MNV or the equivalent virus carrying the FLAG epitope purification tag in either NS1/2 or NS4, and the viral replication complex was purified. The experiment was performed three times by swapping the labelled derivatives of arginine and lysine as described in the Materials and methods. Silver stain of the purified complexes confirmed the presence of the bait proteins, with both the uncleaved and cleaved forms of NS1/2 and NS2 being highly enriched (*Figure 2B*). As expected, complexes purified from NS1/2-Flag virus infected cells co-purified untagged NS4 and vice versa (*Figure 2B*), as we have previously shown these proteins to interact to form a complex (*Thorne et al., 2012*). Western blot analysis of the purified complexes confirmed that viral non-structural and structural proteins were specifically enriched in the purified complexes, including NS5 (VPg)-containing precursors (*Figure 2C*). We noted that anti-NS4 monoclonal antibody was unable to detect protein in the extracts prior to enrichment, which most likely reflected the limited sensitivity of the antibody. Quantitative mass spectrometry of the purified complexes allowed the identified of viral and cellular proteins enriched in the complex (*Figure 2D* and *Supplementary file 2*).

As expected, all viral proteins, including the VF1 protein product of ORF4, an innate immune antagonist (*McFadden et al., 2011*), were enriched in the viral replication complex. There was a significant correlation between the relative enrichment of proteins identified using NS1/2 and NS4 (Spearman correlation of 0.8832), fitting with our prior knowledge that both proteins form a complex during viral replication (*Thorne et al., 2012*). Ontology analysis indicated that proteins involved in vesicle transport and fatty acid metabolism were significantly enriched (*Figure 2—figure supplement 1* and *Supplementary file 3*), fitting with previous observations that the viral replication complex is associated with cytoplasmic membranous structures (*Cotton et al., 2017*; *Hyde and Mackenzie, 2010*; *Hyde et al., 2009*). Several host proteins previously identified in a variety of biochemical and genetic screens were enriched (*Figure 2—figure supplement 1* and *Supplementary file 3*) providing additional confidence that the approach identified biologically relevant interactions. We noted that the VapA and the paralogue VapB, which we have recently identified as binding to the NS1/2 protein (*McCune et al., 2017*), were both highly enriched.

We observed that NS4 failed to enrich mature VPg, instead purifying only the polyprotein cleavage intermediate NS4/5(VPg) (*Figure 2C* and S2.2). In contrast, NS1/2 effectively pulled down mature VPg (*Figure 2C* and S2.2). Consistent with this, NS1/2 enriched the VPg binding partner eIF4G more than 2-fold, while pulldown of NS4 resulted in no eIF4G enrichment, likely indicating that the NS4/5 (VPg) precursor cannot effectively function in translation initiation (*Figure 2—figure supplement 2*). This is consistent with a previous observation that binding of C-terminal VPg fusion proteins to eIF4G is inhibited (*Leen et al., 2016*), but may further suggest that the function of VPg is also altered when present as an N-terminal fusion with NS4. Comparison with the proteomics data obtained using VPg as a bait protein (*Figure 1*) showed some degree of overlap with our replication complex proteome data (*Figure 2—figure supplement 2*). Consistent with our observations with

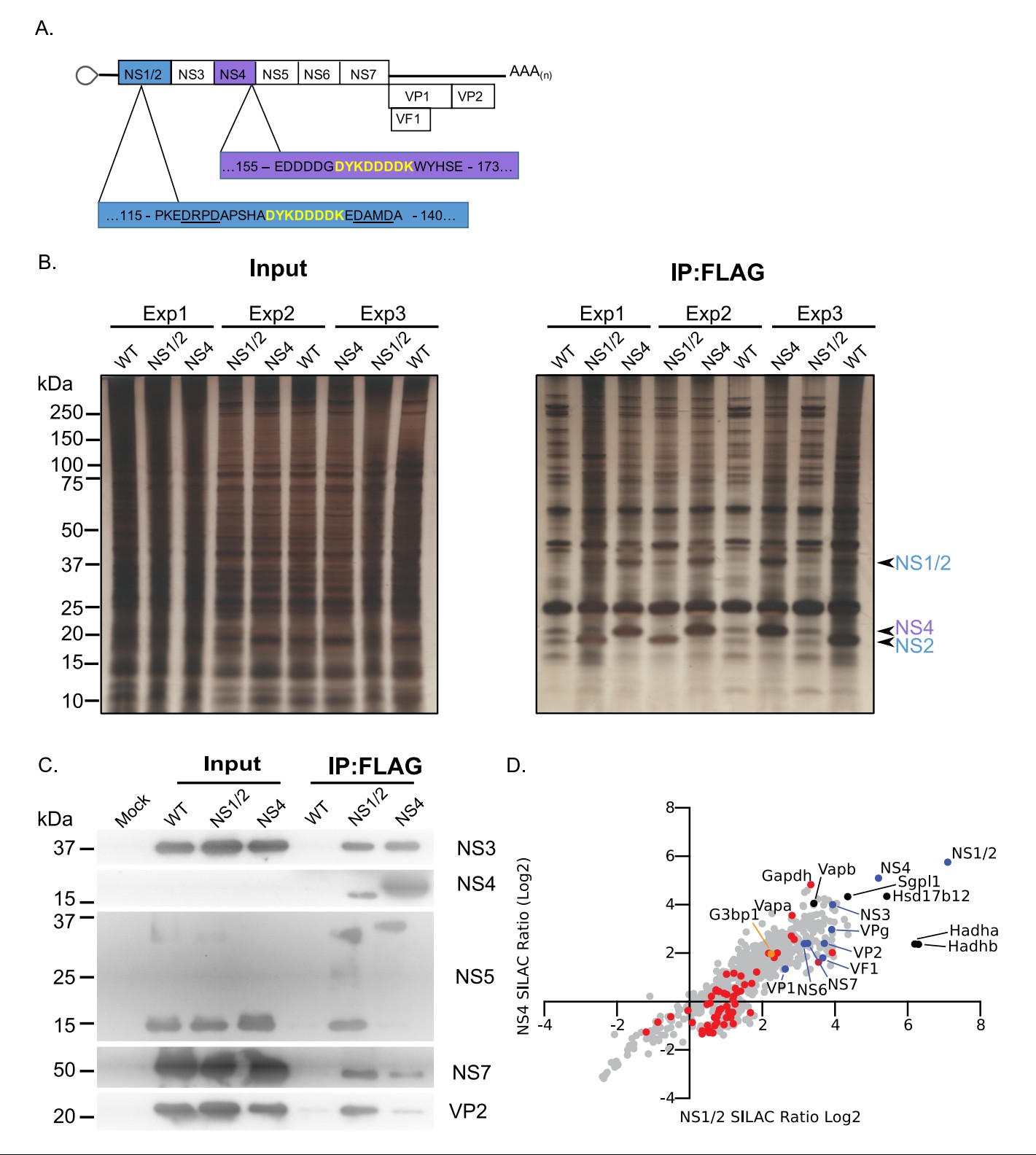

**Figure 2.** Proteomic characterisation of the norovirus replication complex using infectious epitope tagged MNV. (**A**) Schematic representation of NS1/2-FLAG and NS4-FLAG viruses contain insertions of nucleotide sequences encoding the FLAG peptide DYKDDDDK (in yellow) in their coding sequences. The NS1/2-FLAG virus FLAG peptide was inserted between 2 of the three caspase-3 cleavage sites present in NS1/2 (underlined). (**B**) BV2 cells labelled with stable derivatives of arginine and lysine were infected with either wild type MNV (WT) or recombinant epitope-tagged MNV as described in the Materials and methods. 12 hr post infection samples were lysed, samples pooled and immunoaffinity purifications performed as
*Figure 2 continued on next page*

*Figure 2 continued*

described in the text. Samples of the cell lysates (Input) and the affinity purified complexes (IP:Flag) were analysed by SDS-PAGE on a 4–12% gradient gel prior to silver staining. The positions of the NS1/2, NS2 and NS4 proteins is shown. (C) Western blot analysis of lysates purified from cells infected as described in panel B, for various viral proteins, confirming the specific enrichment of viral replicase components. (D) Plot comparing the proteins identified in the complexes purified from cells infected with the NS1/2 and NS4 Flag expressing viruses. The raw data associated with this figure is presented in *Supplementary file 2*. Proteins identified in at least two of the three biological repeats are shown. SILAC ratios were calculated as an average of the two or three biological samples. All MNV proteins were identified in association with NS1/2 and NS4 (light blue) including the viral polymerase NS7, demonstrating enrichment of the MNV replication complex. Proteins previously identified as host factors potentially involved in some aspect of the norovirus life cycle through various biochemical or genetic screens are shown in red. Selected highly enriched proteins are highlighted in black. The NS1/2 binding partner VapA (*McCune et al., 2017*) and paralog VapB were both enriched by NS1/2 and NS4.

DOI: https://doi.org/10.7554/eLife.46681.005

The following figure supplements are available for figure 2:

**Figure supplement 1.** Additional analyses of NS1/2 and NS4-associated proteins.

DOI: https://doi.org/10.7554/eLife.46681.006

**Figure supplement 2.** Further analysis of proteins enriched through MNV VPg proteomics and FLAG-tagged virus replication complex proteome data (Log$_2$ SILAC ratio >1).

DOI: https://doi.org/10.7554/eLife.46681.007

the well-established VPg partner eIF4G, most host factors identified using VPg were enriched by >2 fold using only the NS1/2 tagged virus, and not the NS4-tagged virus. Ontology analysis of host factors enriched by both VPg and NS1/2 more than 2-fold reveals a cross-section of the replication complex proteome dedicated to translation, and RNA metabolism (*Figure 2—figure supplement 2* and *Supplementary file 3*). One protein was enriched by VPg, NS1/2, and NS4 – the core stress granule protein G3BP1 (*Figure 2—figure supplement 2*).

## Identification of host factors required for norovirus infection using a CRISPR- knockout screen

A high density CRISPR library screen was undertaken to identify genes that contribute to the norovirus life cycle. The Brie library (*Doench et al., 2016*) was selected due to the reduced off-target effects relative to previously described CRISPR libraries used for norovirus studies (*Haga et al., 2016*; *Orchard et al., 2016*). In addition, to minimise the impact of gRNAs that may have deleterious effects on long term cell viability and to increase our ability to detect genes that may be important, but not essential, for norovirus-induced cell death, the infection was reduced to 24 hr as compared to 2–10 days post infection in previous studies. BV2-Cas9 expressing cells were infected with lentiviruses carrying the Brie gRNA library carrying 78,637 independent guide RNAs to 19,674 genes (*Doench et al., 2016*). The transduced cells were then infected with two MNV strains, CW3 and CR6, which cause acute and persistent infections in immunocompetent mice respectively (*Nice et al., 2013*; *Thackray et al., 2007*), and guide RNA abundance compared to mock infected cells at 24 hr post infection as illustrated in *Figure 3A*. Genes that were enriched by STARS analysis following MNV infection represent putative pro-viral factors which when disrupted, resulting in slower cell death, whereas those with a negative STARS value represent putative anti-viral factors where virus-induced cell death has occurred quicker, resulting in their underrepresentation in the final pool of cells. MNV-CR6 infection resulted in 212 genes being enriched and 42 being negatively selected (*Figure 3B*), whereas for MNV-CW3 279 and 18 genes were positively and negatively selected respectively (*Figure 3B*). In most cases, there was a clear correlation between the datasets obtained using either strain (*Figure 3C*). STARS analysis was used to ranks genes with positive and negative values with an FDR value less than 0.05 (*Supplementary file 4*). In both screens, the MNV receptor Cd300lf was the most highly positively selected gene identified, in agreement with previous reports (*Haga et al., 2016*; *Orchard et al., 2016*). The previously characterised pro-viral MNV gene VapA (*McCune et al., 2017*) was also identified in both screens (*Supplementary file 4*). The second most highly enriched gene was G3BP1, a gene also identified in one of the two previous CRISPR screens performed on norovirus infected cells (*Orchard et al., 2016*).

Comparing the CRISPR data obtained in this study using two divergent strains of MNV showed a high degree of overlap, with 89 common pro-viral hit and five common anti-viral hits (*Supplementary file 4*). However comparison with the previous reported CRISPR screen used to

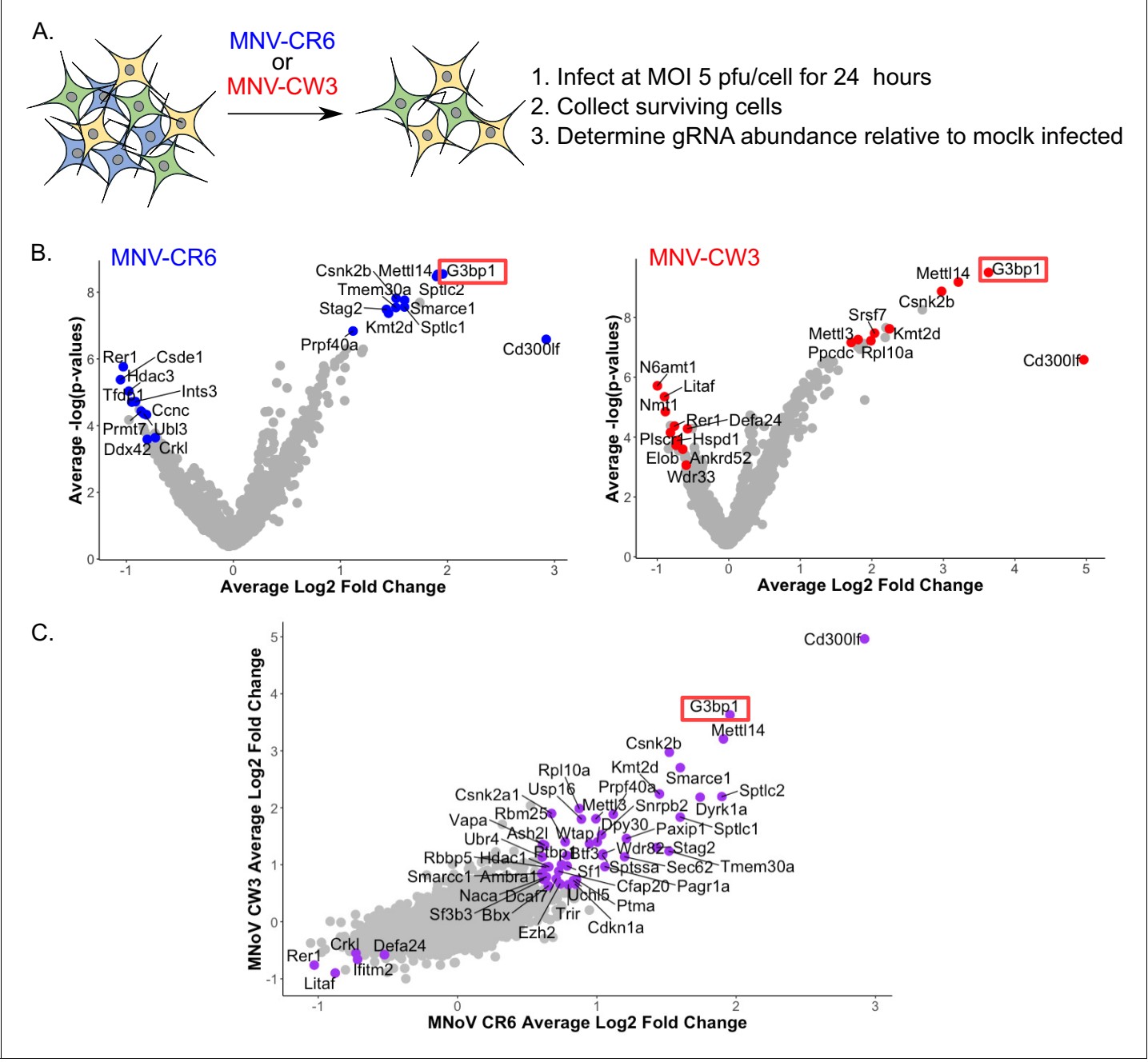

**Figure 3.** CRISPR screen identifies host genes positively and negatively selected upon MNV infection. (**A**) Schematic overview of the infection CRISPR screen workflow. BV2 cells expressing Cas9 were transduced with a CRISPR library then subsequently infected with either MNV CR6 or CW3 for 24 hr. Cells remaining after 24 hr were harvested and used for guide RNA abundance analysis as described in the text. (**B**) Volcano plot identifying candidate genes enriched upon MNV-CW3 (red) or MNV-CR6 (blue); red or blue labelled genes correspond to the top-ten positive or negatively selected genes ranked by the STARS algorithm. (**C**) Plot comparing the Log2 fold change in guide RNA abundance in the CRISPR library transduced BV2 cells following infection with either CR6 or CW3 MNV.
DOI: https://doi.org/10.7554/eLife.46681.008

identify the MNV receptor (*Orchard et al., 2016*) showed only seven proviral hits for CW3 and 8 for CR6, of which only four were common for both strains across all screens (*Supplementary file 4*). The four genes identified were the MNV receptor CD300lf, the stress granule component G3BP1, the histone methyltransferase Kmt2d and Smarce1, which encodes an actin dependent regulator of

chromatin. The discrepancy between the screens performed here and those in previous studies likely reflects the experimental conditions under which the screens were undertaken that is 24 hr of infection vs 2–3 days in the previous screens. The previous screen was performed under much more stringent conditions that were unlikely to identify proteins that play a role in the norovirus life cycle but are not essential. Pathway over-representation analysis (*Liao et al., 2019*) of the genes pro-viral genes identified in the CRISPR screens performed in this study highlighted mRNA processing, RNA splicing and methylation as the three most highly represented gene ontology (GO) terms (*Supplementary file 4*). The gene encoding PTBP1, polypyrimidine tract binding protein, was one of the genes identified in these enriched GO terms. PTBP1 is a protein we have previously shown to bind to a 3′ pyrimidine-rich stretch in the MNV genome which is important for viral pathogenesis (*Bailey et al., 2010*). The interaction of PTBP1 with the viral 3′ end also facilitates the recruitment of PTBP1 to the viral replication complex where it contributes to an as yet undefined aspect of the norovirus life cycle (*Vashist et al., 2012b*).

A cross-comparison of the data obtained from all three approaches allowed us to identify several host proteins that were common to all screens (*Supplementary file 5*). G3BP1, the core stress granule component was identified in all three screens as a potential host factor essential for norovirus infection. G3BP1 was found to be associated with the MNV and NV VPg proteins (*Figure 1D*), enriched in viral replication complexes purified using either NS1/2 or NS4 flag tagged viruses (*Figure 2D*) and identified in a CRISPR screen using two different MNV strains as a putative pro-viral factor involved in the norovirus life cycle (*Figure 3C*).

## G3BP1 is essential for murine norovirus replication

To validate the importance of G3BP1 in the norovirus life cycle we generated G3BP1 deficient BV2 cell lines (*Figure 4A*) and examined the impact of G3BP1 ablation on MNV infection. Western blotting confirmed the loss of G3BP1 in the three lines tested and we noted that at in some cases, a concomitant increase in G3BP2 expression was observed as has been previously noted (*Kedersha et al., 2016*). A clear defect was observed in the ability to replicate to produce infectious virus in three independently selected ΔG3BP1 cell lines (*Figure 4B*). This effect was mirrored by an inability to induce cytopathic effect leading to virus-induced cell death (*Figure 4C and D*). In contrast, the ability of encephalomyocarditis virus (EMCV) to infect and cause cell death was unaffected by the deletion of G3BP1 (*Figure 4C and D*). These data confirm that cells lacking G3BP1 are highly resistant to norovirus infection.

## G3BP1 is essential for human norovirus replication in cell culture

To determine if the G3BP1 was also essential for HuNoV, we examined the impact of loss of G3BP1 on human norovirus replication in cell culture using the Norwalk virus replicon. To establish the experimental system, we first confirmed that the presence of VPg on the 5′ end of the Norwalk RNA was essential for the replication of the replicon RNA and for the capacity to form G418 resistant colonies. Transfection of replicon RNA, purified from replicon containing cells, into BHK cells readily resulted in the formation of antibiotic resistant cell colonies (*Figure 5A*). In contrast, RNA that was proteinase K treated prior to transfection was unable to produce replicon containing colonies. Transfection of replicon RNA into wild type U2OS osteosarcoma cells allowed the formation of replicon-containing colonies, although the efficiency of formation was significantly less than that seen in BHK cells (*Figure 5B*). CRISPR modified U2OS cells that lacked G3BP1 (*Kedersha et al., 2016*) were unable to support NV replication, as evident by the lack of antibiotic resistant colonies (*Figure 5B*). To further examine the role of G3BP1 in human Norwalk virus replication, WT or G3BP1 deficient U2OS cells were transfected with NV replicon VPg-linked RNA, and RNA synthesis monitored overtime following the addition of G418. While a significant increase in NV viral RNA levels was seen in WT U2OS cells, those lacking G3BP1 were completely unable to support NV RNA synthesis (*Figure 5C*). We further validated these observations by transfection of the NV replicon in murine BV2 microglial cells which were able to support HuNoV replication by the formation of small microcolonies of antibiotic resistant cells (*Figure 5D*). In the absence of G3BP1 the formation of antibiotic resistant microcolonies was completely ablated and then subsequently restored in ΔG3BP1 cells engineered to express the wild type version of G3BP1 (*Figure 5D*). These data indicate that like for MNV, G3BP1 is essential for human Norwalk virus replication.

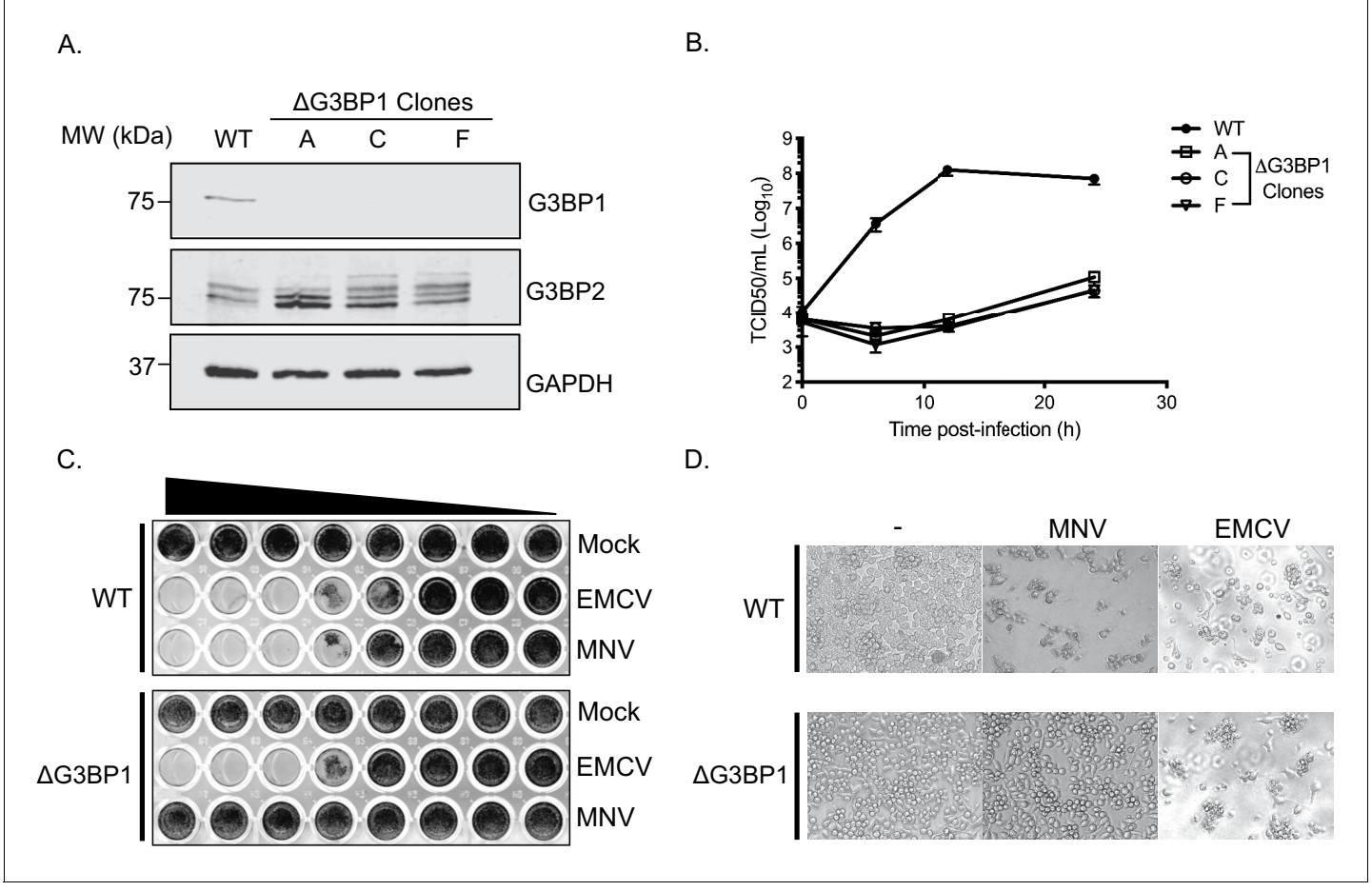

**Figure 4.** CRISPR knockout of G3BP1 renders cells non permissive for MNV replication. (**A**) Western blot analysis of three independent ΔG3BP1 clones for GAPDH, G3BP1 and G3BP2. (**B**) High multiplicity, single cycle growth curve analysis of the impact of G3BP1 ablation on MNV replication. BV2 ΔG3BP1 clone C cells were infected at a MOI of 10 TCID50/cell, samples were collected at the time points illustrated, the samples then processed and titrated by TCID50 as described in the text. The error bars represent standard errors of three biological repeats and the data are representative of at least three independent experiments. (**C**) Wild type (WT) or ΔG3BP1 clone C BV2 cells were plated in a 96 well plate and subsequently infected using a serial dilution of either EMCV or MNV. Cells were fixed in paraformaldehyde and stained with crystal violet 5 days post infection. (**D**) Light micrographs of WT or ΔG3BP1 cells either mock infected (-) or infected with EMCV or MNV and visualised 5 days post infection.
DOI: https://doi.org/10.7554/eLife.46681.009

## The RNA-binding domain of G3BP1 is required for its function in the norovirus life cycle

To confirm the role of G3BP1 in the norovirus life cycle we examined the ability of full length and truncated versions of G3BP1 to restore norovirus replication in G3BP1 knockout cells. A mouse BV2 G3BP1 knockout cell line was complemented with either full length G3BP1 or variants lacking the RGG or both the RGG and RRM binding domains (*Figure 6A*) and the impact on viral replication assessed. Complementation with full length murine G3BP1 restored the ability of MNV to induce cell death (*Figure 6B*) and to produce infectious virus (*Figure 6C*) back to near wild type levels. In contrast, complementation with a variant carrying a deletion of the RGG domain resulted in limited complementation, and deletion of both the RGG and RRM domains together resulted in complete loss of complementation capacity (*Figure 6B and C*). These data confirm that the RNA binding domains of G3BP1 are essential for its function in the norovirus life cycle.

To further define the role of G3BP1 in the norovirus life cycle and to confirm that the function of G3BP1 was downstream of virus binding and viral entry, we therefore bypassed the entry phase of the infection process and transfected MNV VPg-linked RNA into WT and two independently generated BV2 ΔG3BP1 cell lines and examined the impact on norovirus replication. Transfection of MNV

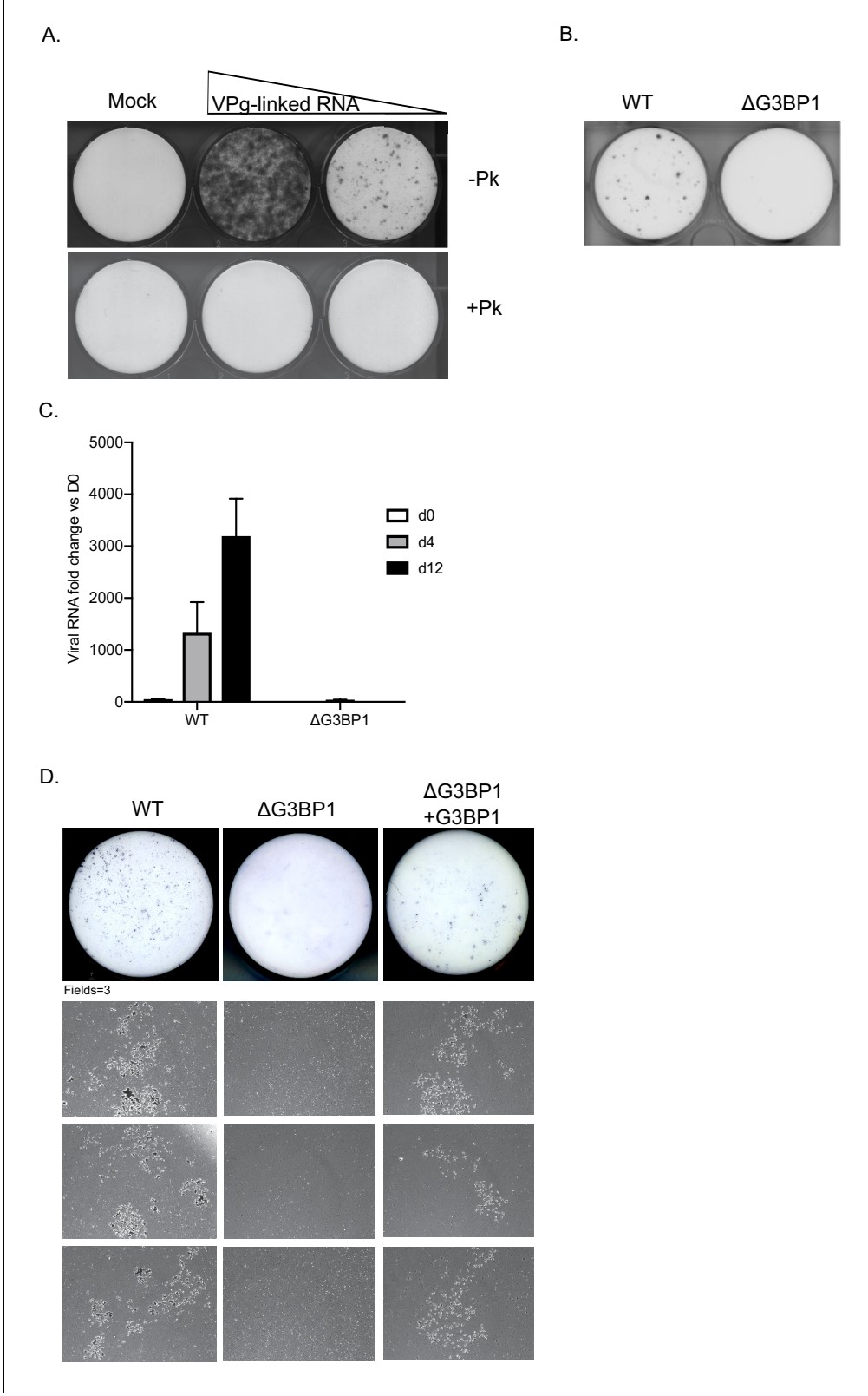

**Figure 5.** G3BP1 is required for human Norwalk virus replication in cell culture. (**A**) Colony formation ability of human norovirus VPg-linked RNA isolated from BHK-NV replicon containing cells is dependent on the presence of VPg. NV VPg-linked RNA isolated from BHK-NV cells was either mock treated or treated with proteinase K prior to transfection into BHK cells. Wells were transfected with either 1.5 μg or 0.75 μg of total RNA isolated from NV replicon containing BHK cells. Following 2 weeks of antibiotic selection with G418, surviving replicon containing colonies were fixed and stained

*Figure 5 continued on next page*

*Figure 5 continued*

with crystal violet in paraformaldehyde. (B) NV replicon colony forming assay in WT and G3BP1-/- U2OS cells performed as described in panel A, with the exception that colonies were stained 12 days post transfection. (C) Quantification of NV replication in WT or ΔG3BP1 U2OS cells following transfection of viral VPg-linked RNA. Viral RNA was quantified by RT-qPCR following transfection and antibiotic selection. The error bars represent the standard error of three biological repeats and are representative of three independent experiments. (D) Analysis of the impact of loss of G3BP1 on NV replication in BV2 cells. Wild type BV2 cells, ΔG3BP1 BV2 cells or ΔG3BP1 BV2 cells reconstituted with wild type full length G3BP1 were transfected with NV replicon VPg-linked RNA. Following selection with G418 for 3 weeks, the cells were fixed and stained with toluidine blue to facilitate the visualisation of microcolonies. Pictures of the entire well and three independent fields of view are shown.

DOI: https://doi.org/10.7554/eLife.46681.010

viral VPg-linked RNA into WT cells resulted in high yields of infectious virus (*Figure 7A*) and viral proteins (*Figure 7C*). The levels of infectivity obtained following transfection of ΔG3BP1 cell lines with MNV viral RNA was comparable to that obtained in WT cells in the presence of the nucleoside analogue 2′-C-methylcytidin (2CMC), a known inhibitor of the norovirus RNA polymerase (*Rocha-Pereira et al., 2012*; *Rocha-Pereira et al., 2013*) (*Figure 7A*). No viral proteins were detected in either of the ΔG3BP1 cell lines suggesting a defect at a very early stage in the viral life cycle (*Figure 7C*). Transfection of VPg-linked RNA into the ΔG3BP1 cell lines reconstituted with WT G3BP1 restored the ability to produce infectious virus (*Figure 7B*) and the production of viral proteins (*Figure 7D*). A minor increase in viral infectivity was observed in the ΔG3BP1 cell line reconstituted with the ΔRGG construct producing viral titres that were higher than those obtained from the WT complemented line in the presence of 2CMC, suggesting low levels of viral replication (*Figure 7B*). However, the levels of viral proteins produced in this line was below the limit of detection by western blot (*Figure 7D*). These data confirm that G3BP1 is required for a post entry stage of the norovirus life cycle and that in the absence of G3BP1 no norovirus replication was observed within the sensitivity of the assay used.

## G3BP1 is required for, or prior to, viral negative strand RNA synthesis

To define the precise role of G3BP1 in the early stages of the virus life cycle, we used strand-specific RT-qPCR to quantify the levels of viral positive and negative sense RNA in WT and ΔG3BP1 cell lines following infection with MNV. As a control, 2CMC was included following virus inoculation as illustrated in the experimental time line (*Figure 8A*). The production of viral positive sense RNA was reduced to background levels in the absence of G3BP1, comparable to levels observed when the 2CMC was present during the infection (*Figure 8B*). Viral negative sense RNA synthesis was also reduced to below the detection limit of the assay in ΔG3BP1 cell lines (*Figure 8C*). Surprisingly, we were able to detect an ~5 fold increase in viral negative sense RNA production at 6 hr post infection of WT cells in the presence of 2CMC, which, given that 2CMC was added after the inoculation phase (*Figure 8B*), likely represents the first round of viral negative sense RNA synthesis, confirming the sensitivity of the assay. Addition of 2CMC during the inoculation phase reduced this background levels (data not shown).

Similar results were obtained following transfection of viral RNA into cells to bypass the entry phase; viral positive and negative sense RNA synthesis was near (or below) the sensitivity of the assay following transfection of viral VPg-linked RNA into two independent ΔG3BP1 cell lines (*Figure 8D and E*). Complementation with WT G3BP1, but not the mutant forms lacking the RNA binding domains, also restored viral positive and negative sense RNA synthesis (*Figure 8F and G*). We did not detect viral positive or negative sense RNAs in the ΔRGG complemented cell line, despite the presence of low levels of viral infectivity (*Figure 7B*). This discrepancy likely reflects the relative sensitivities of the assays and the nature of the strand specific qPCR assay which requires low levels of RNA input to maintain strand specificity. Together these data suggest that the function of G3BP1 is prior to, or at the level of viral negative sense RNA synthesis, with the most logical steps being either viral RNA translation or the formation of viral replication complexes.

## G3BP1 is required for the association of VPg with ribosomal proteins

We have previously shown that norovirus VPg interacts with eIF4G to recruit ribosomal subunits and direct viral translation (*Chaudhry et al., 2006*; *Chung et al., 2014*). The interaction between VPg and eIF4G occurs via a direct interaction between the highly conserved C-terminal region in VPg

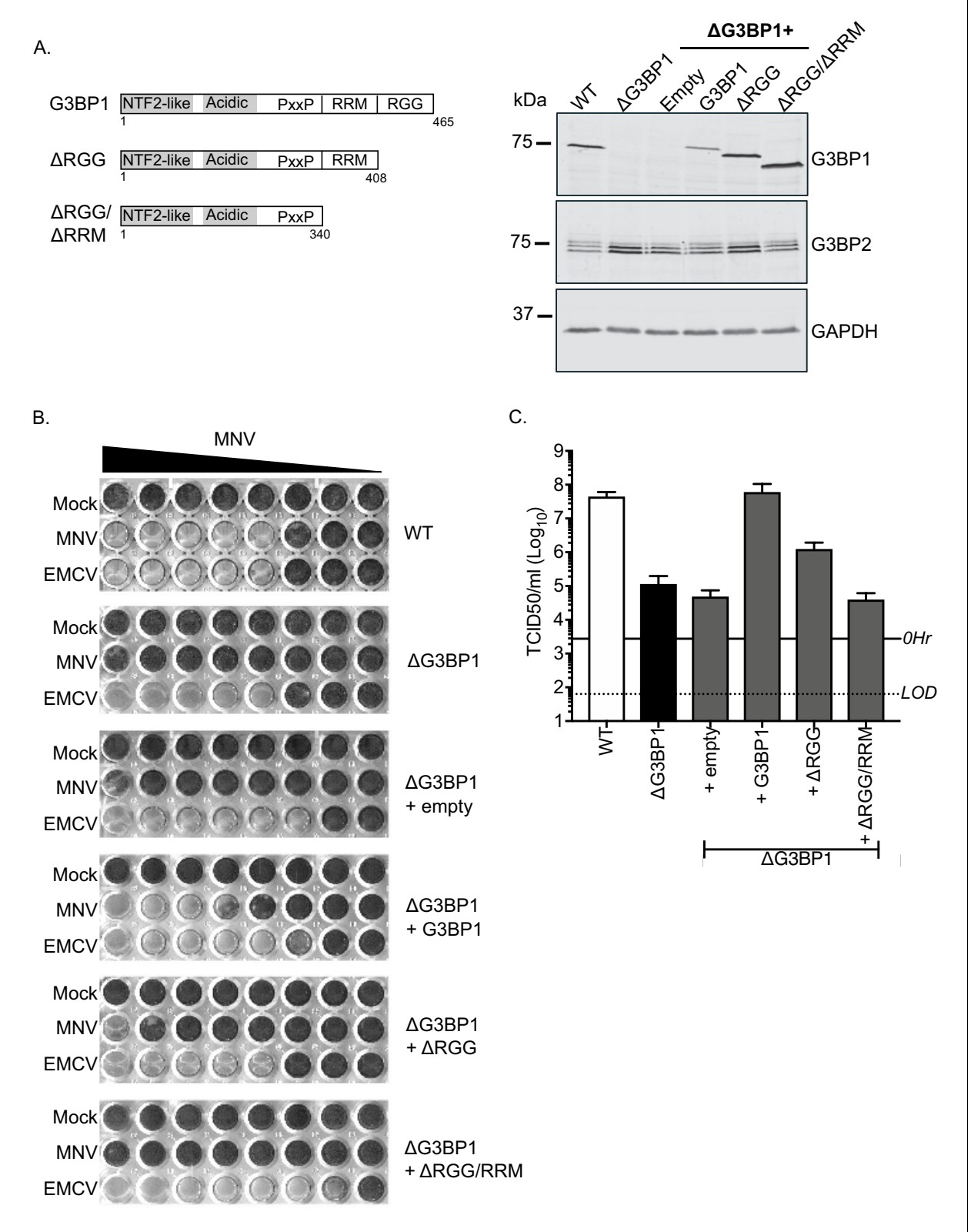

**Figure 6.** MNV replication in BV2 cells requires the RNA binding activity of G3BP1. (**A**) Schematic illustration of the G3BP1 truncations used to identify the domains involved in the norovirus life cycle. The positions of the various domains including the RRM and RGG domains deleted in the ΔRGG and ΔRGG/ΔRRM mutants are also shown. Western blot analysis of wild type BV2 cells (WT) or a ΔG3BP1 cells (clone 1B2) and the respective complemented lines expressing either WT or the various G3BP1 truncations. Cells were lysed prior to separation by 12% SDS-PAGE. (**B**) WT or ΔG3BP1 cells

*Figure 6 continued on next page*

*Figure 6 continued*

complemented with the indicated constructs were plated in a 96 well plate then infected with a serial dilution of MNV, before being fixed and stained 5 days post infection as described in the text. (C) WT or ΔG3BP1 cells complemented with the indicated constructs were infected with MNV at an MOI of 10 TCID50 per cell. After 24 hr the virus yield was determined by TCID50. The error bars represent the standard error of three independent repeats. The data are representative of at least two independently repeated experiments.

DOI: https://doi.org/10.7554/eLife.46681.011

and the central HEAT domain of eIF4G (*Leen et al., 2016*) and does not require any additional cellular cofactors, as a stable complex can be formed between VPg and the eIF4G HEAT domain at least in vitro. The interaction between the eIF4G HEAT domain and the eIF3 complex plays a central role in the recruitment of the 40S ribosomal subunit for translation initiation (*Marcotrigiano et al., 2001*; *Kumar et al., 2016*; *Villa et al., 2013*). Our proteomics analysis also confirms that the norovirus VPg proteins form a complex that contains multiple components of the large and small ribosomal subunits (*Figure 1D*). It has been established previously that G3BP1 associates with 40S subunits (*Kedersha et al., 2016*). To assess a potential role for G3BP1 in the formation of VPg-driven translation complexes in cells, we examined the ability of GFP tagged versions of MNV VPg to pull down ribosomal proteins in the presence and absence of G3BP1. GFP-tagged WT MNV VPg was readily able to pull down eIF4G, G3BP1 and RpS6, a component of the 40S subunit (*Figure 9A*). However, in the absence of G3BP1, the ability to pull down RpS6 and RpS3 was lost (*Figure 9A* and *Figure 9— figure supplement 1*). Furthermore, we found that disruption of the VPg-eIF4G interaction by the introduction of the F123A mutation into the eIF4G binding domain, also significantly reduced the ability to pull down RpS6, RpS3, as well as RpL4, a component of the large ribosomal subunit (*Figure 9B* and *Figure 9—figure supplement 1*). We also observed that upon reconstitution of G3BP1 expression, the levels of RpS6, RpS3 and RpL4 associated with VPg were enhanced (*Figure 9B* and *Figure 9—figure supplement 1*). We note however that the levels of ribosomal proteins associated with the GFP-tagged VPg in this assay is relatively low, we think this reflect the fact that in the context of viral infection where VPg is covalently linked to viral RNA, secondary interactions between translation initiation factors and the viral RNA likely stabilise this complex. We have previously seen that the eIF4A protein binds directly to the 5' end of the sapovirus genome (*Hosmillo et al., 2016*). Furthermore, eIF4G is known to make secondary stabilising interactions with the template RNA that are critical for translation initiation (*Yanagiya et al., 2009*). Therefore we hypothesise that in the absence of covalently linked viral RNA, the association of VPg with ribosomal subunits is less robust and therefore less able to be maintained during the purification process used in the GFP-Trap approach. Therefore, to assess how the loss of G3BP1 may influence the association of VPg-linked viral RNA with ribosomal subunits in a more physiologically relevant system, we quantified the amount of viral VPg-linked RNA bound to ribosomal subunits by RNA-IP. By inclusion of the viral RNA polymerase inhibitor 2CMC we were able to assess viral RNA association with ribosomal proteins in the absence of viral RNA synthesis. We found that in the absence of G3BP1, the amount of viral RNA associated with the ribosomal subunit protein RpS3 was decreased (*Figure 9C*). These data suggest that that G3BP1 likely contributes in some manner to the association of VPg and viral VPg-linked RNA with ribosomal subunits.

## G3BP1 is required for efficient polysome loading of norovirus VPg-linked RNA

To assess the impact of G3BP on the translation of viral VPg-linked RNA following viral infection, we evaluated the impact of loss of G3BP1 on the recruitment of viral RNA to polysomes under conditions where viral RNA synthesis was inhibited, namely in the presence of 2CMC. This approach enabled us to assess only the capacity of the incoming parental viral RNA to assemble into translationally active complexes, a stage often referred to as the 'maiden round' of RNA virus genome translation. To this aim, cells were infected with MNV in the presence of 2CMC and polysomes profiling on extracts prepared from cells at 4 and 9 hr post infection performed (*Figure 10A*). Quantification of the viral RNA levels in cells in the presence of 2CMC confirmed that the absence of G3BP1 has no impact on the overall levels present at the time points examined (data not shown). We noted that even in the presence of 2CMC, which inhibits viral RNA synthesis, there was a small but

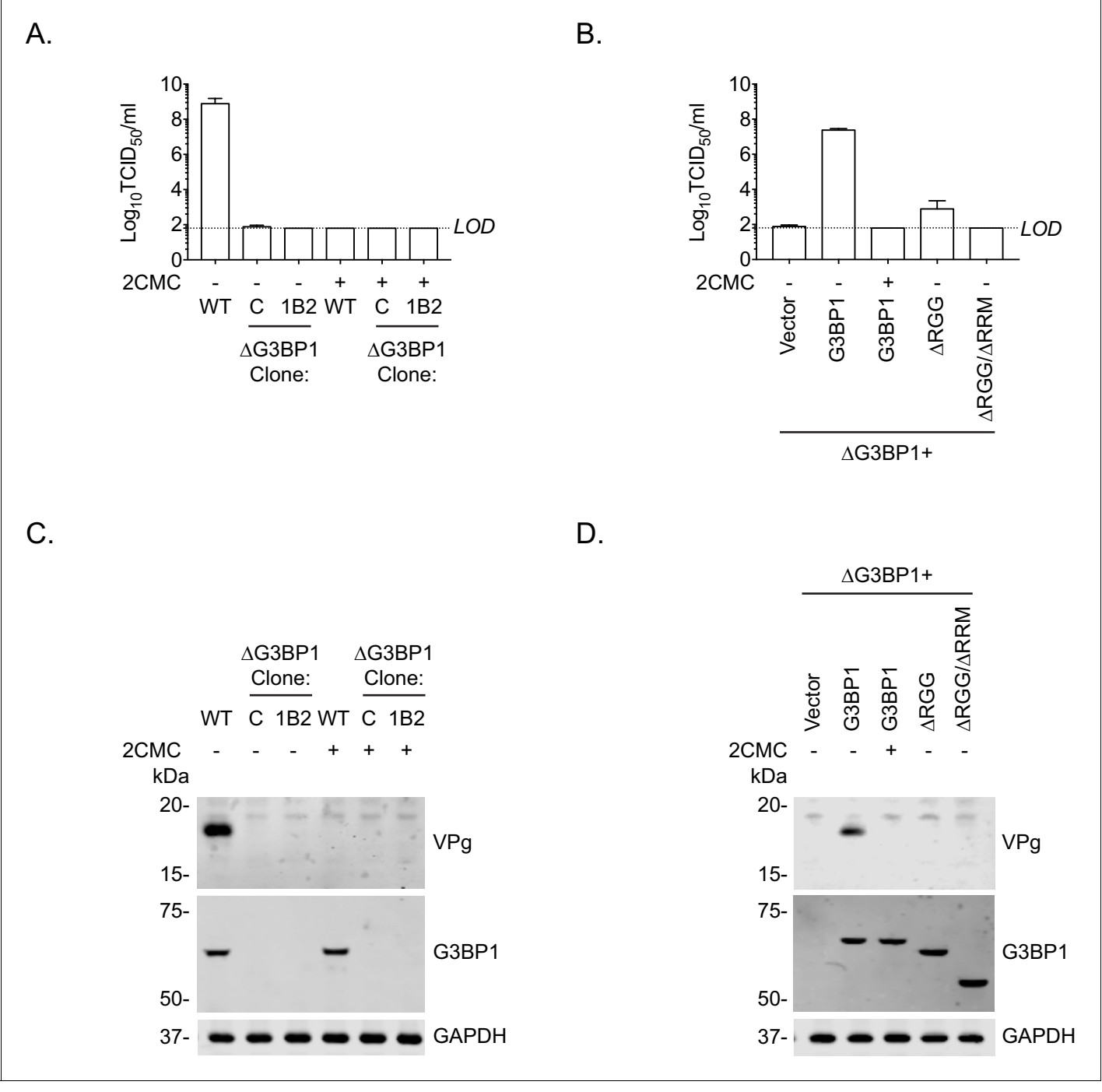

**Figure 7.** Loss of G3BP1 results in a defect following transfection of viral VPg-linked RNA into ΔG3BP1 cells. (**A**) The indicated cell lines were transfected with MNV viral RNA and harvested at 9 hr post transfection for TCID50 to assess the virus yield. In some instances, the nucleoside analogue 2CMC was included to inhibit viral replication. The dotted line indicates the limit of detection (LOD) and the error bars represent the standard error from three biological repeats. (**B**) Infectious virus yield from ΔG3BP1 and reconstituted cell lines performed as described in panel A. (**C**) and (**D**) illustrate the accompanying western blots for samples prepared in panel A and B respectively. Samples were prepared at 24 hr post transfection, prior to harvesting, separation by SDS-PAGE on a 4–12% gradient gel prior to western blotting for the indicated proteins.
DOI: https://doi.org/10.7554/eLife.46681.012

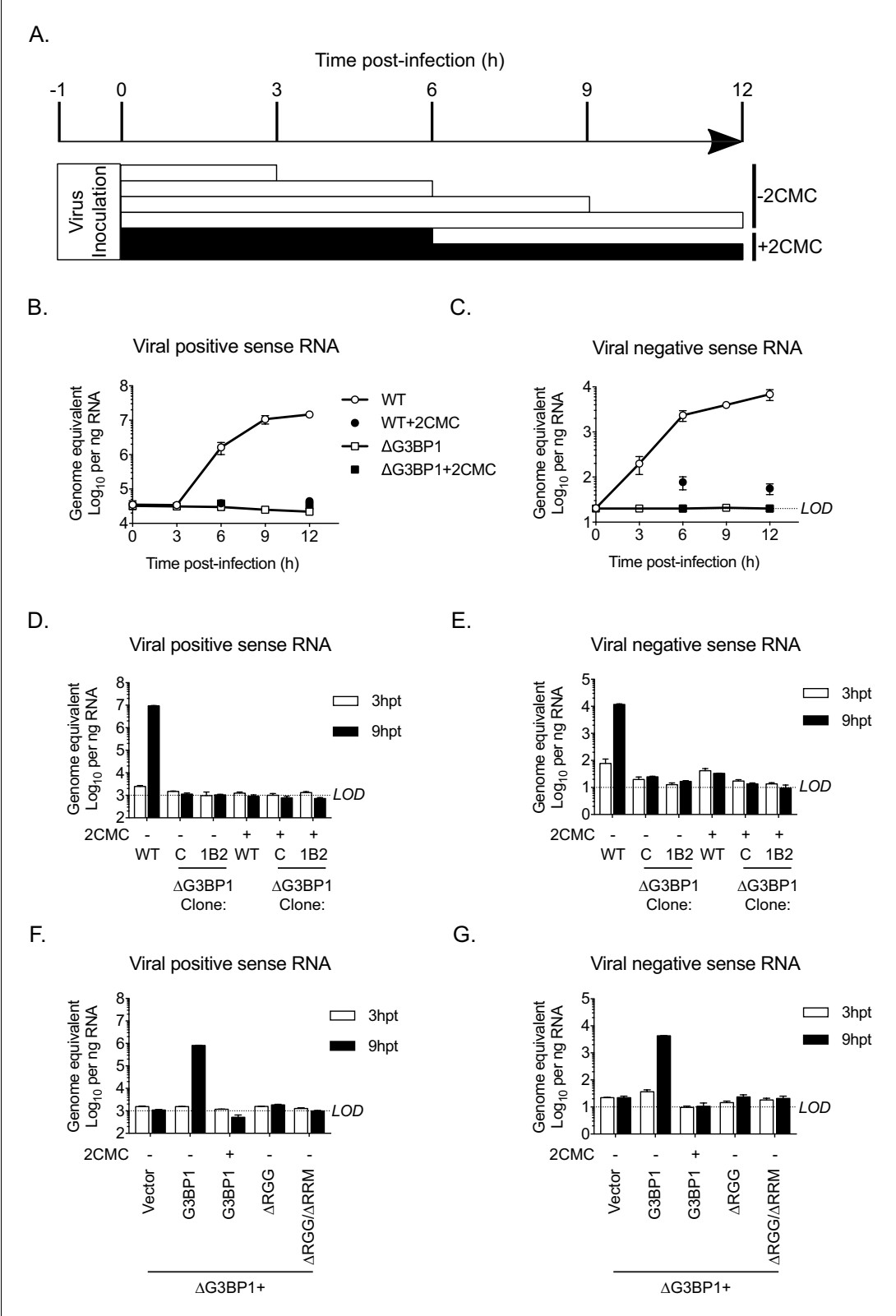

**Figure 8.** The Lack of G3BP1 results in a failure to produce viral negative sense RNA. The experimental design is illustrated in (**A**). Wild type or ΔG3BP1 (1B2) cells were infected prior to the addition of the nucleoside analogue 2CMC to prevent viral RNA synthesis. Samples were harvest at the indicated time post infection and viral positive (**B**) and negative sense RNA quantified by stand specific RT-qPCR (**C**). Error bars represent standard error of three biological repeats. LOD refers to the limit of detection of the assay. (**D**) and (**E**) Viral RNA synthesis following transfection of viral VPg-linked RNA into

*Figure 8 continued on next page*

*Figure 8 continued*

WT or two ΔG3BP1 cell lines. (**F**) and (**G**) Viral RNA synthesis following transfection of viral VPg-linked RNA into ΔG3BP1 (1B2) complemented with full length G3BP1 or truncated derivatives. Error bars represent standard error of three biological repeats.

DOI: https://doi.org/10.7554/eLife.46681.013

measurable increase in free 80S ribosomes over time in WT cells but not in cells lacking G3BP1 (*Figure 10A*). We have previously found that MNV infection results in translation shut off and that this effect is at least partially due to the activity of the NS6 protease (*Emmott et al., 2017*). The fact we observed 80S accumulation in WT cells, even in the absence of viral RNA synthesis, but not in cells lacking G3BP1, indirectly lead us to suspect that translation of viral RNA had occurred in WT cells, but was much less efficient in cells lacking G3BP1. Further analyses indicated that while most ribosome-associated norovirus RNA in WT cells was found in polysomes containing fractions, less viral RNA was found in ribosome-containing fractions (1–12 in *Figure 10A and B*) in the absence of G3BP1 and, in comparison to WT cells, very little viral RNA was found in fractions containing polysomes (*Figure 10B*). Extending the fractionation to include the free RNA and ribonucleoprotein complexes at the top of each gradient confirmed that in the absence of G3BP1 norovirus RNA is less efficient at assembling into polysomal fractions, suggesting a defect at the level of viral protein synthesis (*Figure 10C*). Together these data support the hypothesis that G3BP1 functions to promote the translation of norovirus VPg-linked RNA, by facilitating the association with ribosomal subunits and the formation of polysomes on viral RNA.

## G3BP1 is require for efficient norovirus VPg-dependent translation

To further examine a potential role of G3BP1 in norovirus VPg-dependent translation, cytoplasmic translationally competent extracts were prepared from WT and ΔG3BP1 cell lines. In order to ensure comparable overall translation efficiencies between extracts, RNA generated from a cricket paralysis virus IRES (CrPV) bicistronic reporter plasmid (a kind gift from Professor Ian Brierley, University of Cambridge) was used to measure cap-dependent and cap-independent translation (*Wang and Jan, 2014*). The dual luciferase construct was in vitro transcribed, capped and poly(A) tailed. Renilla luciferase was produced via cap-dependent translation initiation whilst firefly luciferase was synthesised via cap-independent, CrPV IRES-dependent translation (*Figure 11A*). Both cap-dependent (*Figure 11B*) and CrPV IRES-dependent translation (*Figure 11C*) were comparable in extracts prepared from WT and ΔG3BP1 cell lines (*Figure 11A*). To compare norovirus VPg-dependent translation efficiency between WT and ΔG3BP1 lysates, viral VPg-linked RNA was first extracted from sucrose cushion-purified MNV virions and the presence of VPg on the viral RNA 5' end confirmed by resistance to XrnI mediated degradation in vitro (*Figure 11—figure supplement 1* panel A). The translation profile of the purified viral RNA was further analysed by translation in rabbit reticulocyte lysates in comparison to in vitro transcribed viral genomic and sub-genomic RNAs (*Figure 11—figure supplement 1* panel B). We found that norovirus VPg-dependent translation was reduced in nuclease treated extracts prepared from cells lacking G3BP (*Figure 11D*). Quantification of the levels of multiple viral proteins produced over multiple experiments indicated that translation in nuclease treated extracts was on average reduced by ~40–50% because of G3BP1 ablation (*Figure 11E*). A similar reduction in in vitro translation was observed across multiple time points (*Figure 11—figure supplement 1* panel C). This 50% reduction in translation efficiency was also consistently observed in extracts that were not nuclease treated, and therefore contained physiologically relevant levels of cellular mRNAs (*Figure 11F and G*). These data further confirm that G3BP1 functions to enhance norovirus VPg-dependent translation initiation.

## Discussion

In this study, we have used a combination of biochemical and genetic approaches to identify host factors involved in the norovirus life cycle. Our combined approaches resulted in the identification of the core stress granule component G3BP1 as a host protein critical for the replication of both murine and human noroviruses in cell culture. Furthermore, we determined that G3BP1 plays a key role in

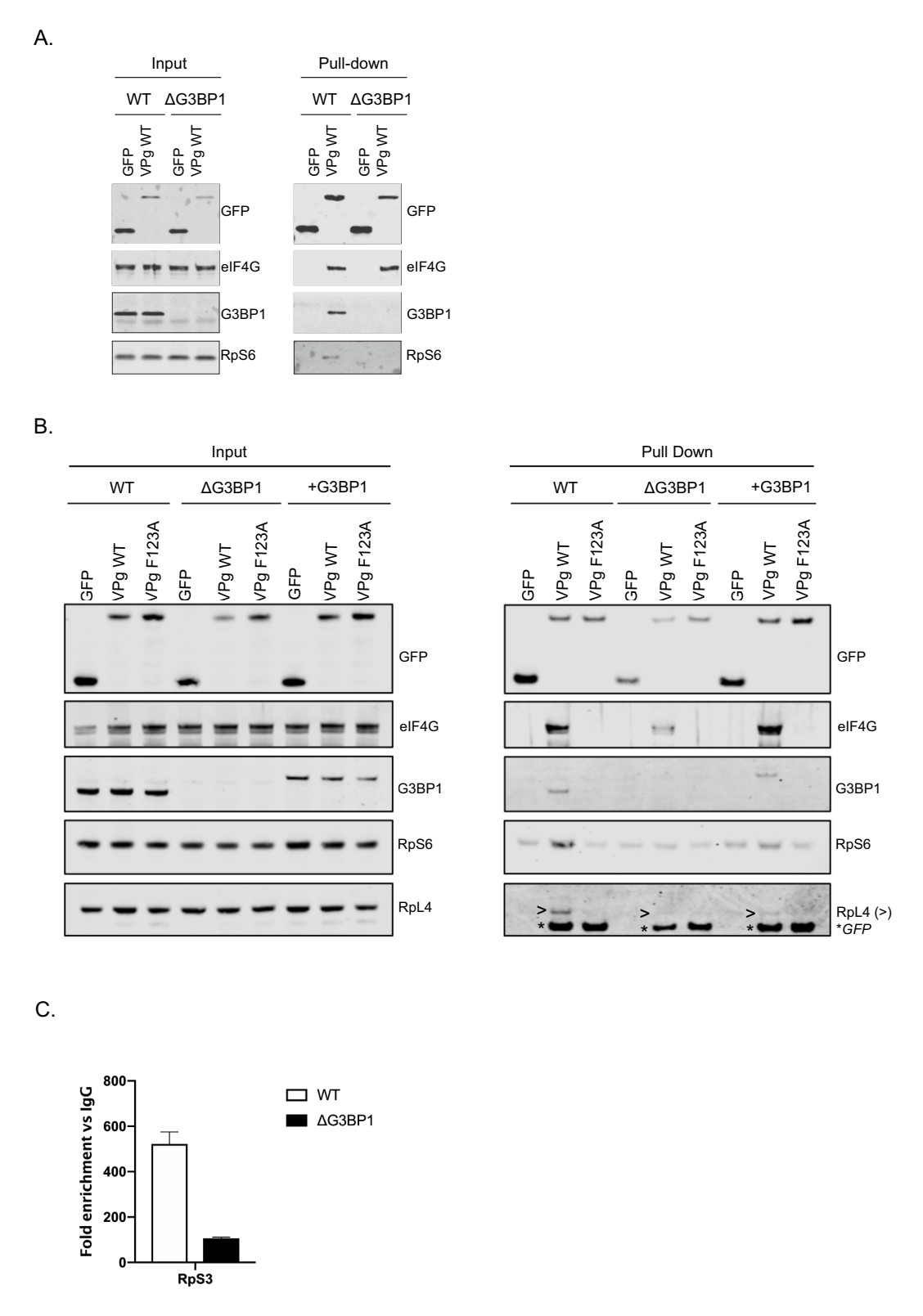

**Figure 9.** G3BP1 is required for the association of VPg and norovirus VPg-linked RNA with ribosomal subunits. (**A**) GFP-Trap immunoprecipitation of complexes isolated on with GFP alone or GFP tagged wild type MNV-VPg demonstrating the pull down of eIF4G1, G3BP1 and 40S subunits (RpS6). BV2 cells were transfected with the relevant constructs, lysates prepared and GFP-Trap pull downs performed as detailed in the text. Samples were separated by SDS-PAGE and western blotted for the proteins as shown. (**B**) Mutations in the eIF4G binding domain ablate the association of VPg with

*Figure 9 continued on next page*

*Figure 9 continued*

G3BP1 and ribosomal subunits and the reconstitution of G3BP1 expression in ΔG3BP1 BV2 cells restores the interaction. GFP-Trap pull downs were performed as described in panel A with the addition of the MNV VPg F123A mutation known to reduce the association with eIF4G and the inclusion of the in ΔG3BP1 1B2 cell line engineered to express a Flag-tagged derivative of G3BP1. Following the pull down samples were analysed by western blot for presence of the ribosomal subunits using RpS6 and RpL4 as markers for the small and large ribosomal subunits respectively. >denoted the RpL4 proteins; whereas, * indicated the presence of GFP-VPg fusion proteins, present on the same membrane due to the sequential probing of the membrane. (C) Loss of G3BP1 results in decreased association between norovirus RNA and ribosomal proteins. WT or cells lacking G3BP1 were infected with MNV in the presence of the RNA polymerase inhibitor 2CMC and the amount of viral RNA found associated with RpS3 determined by RNA-IP. Error bars represent SEM of duplicate samples and the values are expressed as fold enrichment with respect to the IgG control antibody.

DOI: https://doi.org/10.7554/eLife.46681.014

The following figure supplement is available for figure 9:

**Figure supplement 1.** G3BP1 is required for the association of VPg with ribosomal subunits.

DOI: https://doi.org/10.7554/eLife.46681.015

the processes of norovirus VPg-dependent protein synthesis, uncovering a new function for G3BP1 in facilitating RNA virus genome translation.

The orthogonal approaches used in the current study provide an unprecedented insight into the identity of host factors with potential roles in the norovirus life cycle. The detailed proteomic analysis of the viral replication and translation complexes formed during MNV infection (*Figure 2*) resulted in the identification of several host factors with previously identified roles in the MNV life cycle. We focused our efforts on G3BP1 as it was identified in all three approaches and was also identified in a CRISPR screen published during this study (*Orchard et al., 2016*). Furthermore, we have previously shown that feline calicivirus (FCV), a relative of noroviruses within the *Vesivirus* genus, cleaves G3BP1 to inhibit stress granule formation (*Humoud et al., 2016*). In contrast, MNV infection does not result in G3BP1 cleavage and instead forms cytoplasmic foci the composition of which is distinct from canonical stress granules (*Brocard et al., 2018*).

G3BP1 is one member of a group of G3BP proteins (Ras-GTPase-activating protein (SH3 domain)-binding proteins), referred to as Rasputin in insects, that possess RNA binding activity and have multiple cellular functions including the regulation of RNA stability and translation in response to stress. Originally identified as a protein that interacted with Ras-GTPase activating protein (RasGAP), more than two decades of research have significantly expanded our knowledge of the multifunctional role in cellular processes. It is now well accepted that G3BPs play a role in cancer cell survival, cancer metastasis and invasion, processing of specific miRNAs and stress granule formation (Reviewed in *Alam and Kennedy, 2019*). Stress granules are dynamic cytoplasmic ribonucleoprotein complexes that form rapidly under stress conditions and within which cellular RNAs are stored in stalled translation complexes (*Protter and Parker, 2016*). In the context of viral infection, numerous studies have suggested that many, if not all, viruses must interact in some manner with stress granules as there is growing evidence that the formation of cytoplasmic stress granules is part of the anti-viral defense mechanism (Reviewed in *McCormick and Khaperskyy, 2017*). Some viruses interact with stress granules to promote viral replication (*Cristea et al., 2010*; *Kim et al., 2016*; *Panas et al., 2014*; *Panas et al., 2012*) whereas some do so to counteract the inhibitory effect of stress granules on the translation of viral RNA (*Panas et al., 2015*; *White et al., 2007*).

Our data suggests that G3BP1 plays a key role in promoting the translation of norovirus VPg-linked viral RNA. Positive sense RNA viruses have evolved mechanisms to ensure the efficient translation of their viral genomic RNAs in the presence of high concentrations of competing cellular RNAs. These mechanisms include the use of internal ribosome entry site elements (IRES), modified cap-dependent mechanisms (*Firth and Brierley, 2012*; *Jaafar and Kieft, 2019*) and the ability to target the host cell translation machinery to generate an environment where viral RNA translation is favoured over cellular capped RNAs (*Walsh et al., 2013*). G3BP1 is thought to associate primarily with free 40S subunits (*Kedersha et al., 2016*). Our data supports a hypothesis whereby the association of G3BP1 with 40S ribosomal subunits somehow stabilises the recruitment of a translation initiation complex to the 5' end of the VPg-linked viral RNA genome, promoting VPg-dependent translation and thereby uncovering a new function in virus specific translation. The mechanism by which G3BP1 contributes to this process has yet to be fully explored but our data supports the hypothesis that G3BP1 directly or indirectly promotes the recruitment of ribosomal subunits to VPg-driven

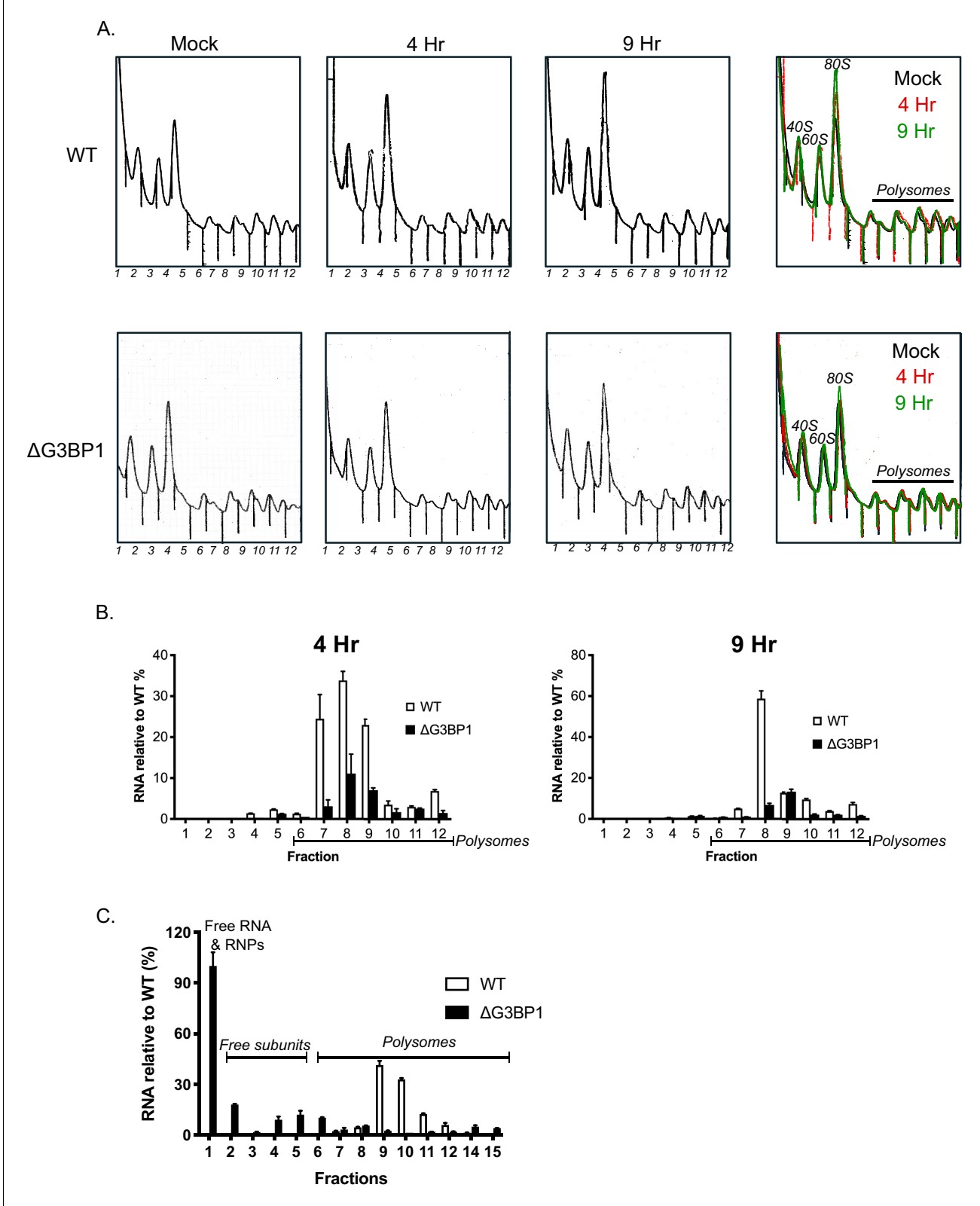

**Figure 10.** G3BP1 is required for polysome association of viral RNA association. (**A**) Polysome profiles of the ribosome containing fractions from mock or MNV infected wild type (WT) or ΔG3BP1 (1B2) BV2 cells at 4 and 9 hr post infection (moi 3 TCID50/cell). (**B**) Relative viral RNA levels present in ribosome containing fractions expressed relative to WT infected BV2 cells. (**C**) Extended gradient fractionation of WT or ΔG3BP1 cells infected with MNV and harvested 9 hr post infection. Viral RNA levels across the gradient are expressed as described in panel B. Error bars represent standard error

*Figure 10 continued on next page*

*Figure 10 continued*

of three technical repeats from each biological samples, defined as a single fraction from a single experiment. The data shown in panels A-C are representative of at least three experimental repeats.

DOI: https://doi.org/10.7554/eLife.46681.016

translation complexes. The RGG motif of G3BP1 is known to be essential for the association between G3BP1 and 40S subunits as well as the ability to from stress granules, whereas data would suggest that the RRM may play a regulatory role (*Kedersha et al., 2016*). These domains were also required for the function of G3BP1 in the norovirus life cycle (*Figure 6*) confirming that the G3BP1 association with 40S is important for its role in promoting norovirus VPg-dependent translation. Importantly, RGG domains are known to have many functions (*Thandapani et al., 2013*) and therefore in the context of G3BP1 function in the norovirus life cycle, may also contribute to unknown interactions that promote norovirus translation. Previous work on alphaviruses have shown that G3BP1 is sequestered by binding to the nsP3 protein (*Panas et al., 2012*; *Panas et al., 2014*; *Panas et al., 2015*). Furthermore, this interaction occurs via an FGDF motif also found in other viral proteins including the ICP8 protein of herpes simplex virus (*Panas et al., 2015*). While the MNV VPg protein has a similar motif FGDGF (*Figure 1A*), this motif is not conserved in the GI Norwalk virus VPg protein. Therefore, our data suggest that the interaction of VPg with G3BP1 is not direct, fitting with our observation that this interaction is reduced by mutations in the eIF4G binding domain (*Figure 1A* and *Figure 9B*.) While our data fit with a primary role for G3BP1 in norovirus translation, we are unable to exclude the possibility that G3BP1 plays other roles in the viral life cycle. Recent studies have confirmed that G3BP1 is enriched at sites of viral RNA synthesis (*Brocard et al., 2018*; *Fritzlar et al., 2019*) so it is possible that G3BP1 makes multiple contacts between the 40S subunit and viral RNA genome directly. These additional contacts, may further promote viral VPg-dependent translation and/or another aspect of the viral life cycle.

The technical challenges associated with studying human norovirus replication in cell culture have limited the experimental approaches we could use to validate the role of G3BP1 in human norovirus translation. However, our results have clearly demonstrated that in the absence of G3BP1, human Norwalk virus is unable to replicate or form replicon-containing colonies. Furthermore we were able to show that BV2 murine microglial cells support the replication of the HuNoV GI replicon, albeit it to a lesser degree than BHK or U20S cells (*Figure 5*). These data confirm that all the machinery necessary for HuNoV replication is conserved between human, hamster and mouse cells. Furthermore, reconstitution of the ΔG3BP1 BV2 microglial cells with WT G3BP1, at least partially restored the ability of the HuNoV GI replicon to form colonies, confirming the specificity of the effect. The presence of G3BP1 in the NV VPg-containing complexes again fits with our hypothesis that G3BP1 plays a role in promoting viral VPg-dependent protein synthesis.

We have previously found that norovirus infection leads to preferential viral-translation whereby cellular mRNAs induced in response to norovirus infection are inefficiently translated (*Emmott et al., 2017*). This modification of host cell translation is at least partially driven by the ability of the viral NS6 protease to cleave PABP and the induction of apoptosis which results in cleavage of cellular translation initiation factors (*Emmott et al., 2017*). Importantly, whilst caspase cleaved translation initiation factors do not support host cell cap-dependent translation, we have previously found that the cleaved forms of eIF4G do support norovirus VPg-dependent translation (*Emmott et al., 2017*). The ability of the norovirus protease to cleave PABP and other substrates is also controlled by polyprotein processing and interactions with other viral proteins (*Emmott et al., 2019*). Recent work confirms that the preferential viral translation is not driven by the GCN2-mediated phosphorylation of eIF2α in MNV infected cells (*Brocard et al., 2018*). We note however that others have suggested that NS3 may contribute to the translational shut off seen in MNV infected cells (*Fritzlar et al., 2019*), with the caveat that this observation was made outside of the context of infected cells and used overexpressed tagged viral proteins. We suspect that noroviruses use multiple mechanisms that work cooperatively to enable the control of host gene expression and the subsequent translation of the cellular mRNAs. The relative contribution of these processes in any given cell type may also vary dependent on the degree to which the cells can sense and respond to viral infection through the induction of innate and apoptotic responses.

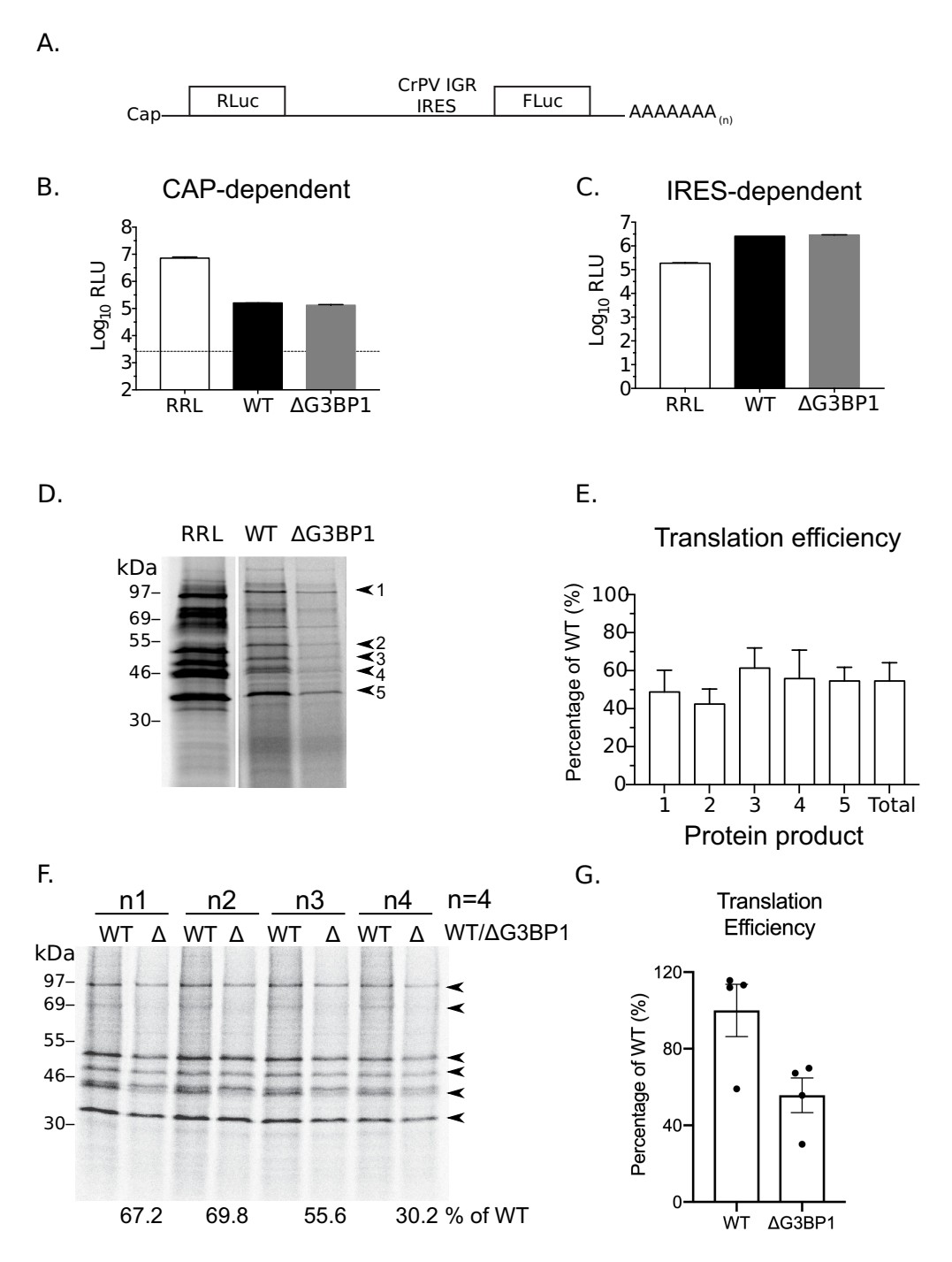

**Figure 11.** G3BP1 is required for efficient norovirus VPg-dependent translation. (**A**) Illustration of the bicistronic construct used to assess the in vitro translation efficiency of the extracts prepared from WT BV2 cells or cells lacking G3BP1. The location of the 5' cap and the 3' poly (A) tails are highlighted in relation to the renilla and firefly luciferase coding regions, along with with the CrPV IRES. Comparsion of cap (**B**) and CrPV-IRES dependent translation (**C**) in translation competent extracts prepared from WT BV2 cells or BV2 cells lacking ΔG3BP1. Extracts were programmed with in vitro transcribed RNA as described in the text and the levels of luciferase compared to those obtained using rabbit reticulocyte lysate (RRL). (**D**) Translation of MNV VPg-linked viral RNA is diminished in nuclease treated extracts prepared from ΔG3BP1 cells. Translation of viral RNA in rabbit reticulocyte lysates (RRL) was used as a side by side comparison. The positions of the viral proteins quantified by phosphor imaging are indicated with arrow heads (1-5). (**E**) Quantification of norovirus protein synthesis for each of the identified protein products in panel D and total translation levels across multiple experiments. The levels of viral translation is expressed as a percentage relative to the same protein product translated in extracts prepared from WT cells. The error bars represent the standard error of three independent experiments. (**F**) Norovirus VPg-dependent translation in

*Figure 11 continued on next page*

*Figure 11 continued*

non-nuclease treated extracts prepared from WT or ΔG3BP1 cells. The total translation efficiency for the viral proteins, highlighted by arrowheads, is shown below the respective lane. Translation efficiency is expressed as a percentage in comparison to the average of translation seen in extracts prepared from WT cells across all biological repeats. E) Quantification of viral proteins produced in panel (D) plotted as % translation efficiency with respect to the translation levels observed in non-nuclease treated extracts from WT BV2 cells.

DOI: https://doi.org/10.7554/eLife.46681.017

The following figure supplement is available for figure 11:

**Figure supplement 1.** Lack of G3BP-1 results in reduced Norovirus VPg-dependent translation efficiency in vitro.

DOI: https://doi.org/10.7554/eLife.46681.018

The observation that many of the factors enriched using the VPg protein were only enriched on complexes purified with NS1/2 tagged infectious MNV, could suggest that the viral proteins present in the viral translation complex are distinct from those present in complexes active for viral RNA synthesis. However, we cannot formally rule out other possible explanations including the possibility that the specific enrichment of translation factors on NS1/2 occurs because NS1/2 is the first protein to be translated from ORF1, therefore unprocessed NS1/2 at the N-terminus of the ORF1 polyprotein being actively translated could function as an anchor, facilitating the enrichment of ribosomes and the associated factors. In addition, we have previously seen that VPg-containing precursors may bind the translation initiation factor eIF4G less well (*Leen et al., 2016*), which could prevent some VPg (NS5) containing precursors associating with translation initiation complexes. Furthermore, the norovirus replication complex contains a number of highly conserved viral protein-viral protein interactions (*Emmott et al., 2019*). Within this network of interactions, the NS1/2 protein interacts with the viral RNA polymerase NS7, which in turn binds VPg. It is therefore possible that the interactions observed between the NS1/2 proteins and the host proteins involved in translation, occurs via the formation of a NS1/2-NS7-VPg complex. How the viral protein-protein interactions within the replication complex integrate with the network of cellular proteins identified in the current study and how this facilitates viral replication and/or regulates the host response to infection, remains to be determined.

This study provides an unprecedented insight into the identity of host factors likely involved in the norovirus life cycle. As detailed above, we focused our detailed analysis on the role of G3BP1 given that it was identified in all three screens, however it is likely that many of the host factors identified in each screen play key roles in the norovirus life cycle. The high enrichment of proteins involved in vesicle-mediated transport, organelle organisation and exocytosis, fits well with our current understanding on the nature of the norovirus replication complex and the impact on host cell processes. Detailed analysis of the impact of norovirus non-structural protein expression and MNV infection on host cell membrane architecture suggests that norovirus replication complexes are generated from the endoplasmic reticulum (ER), and results in the formation of single membrane vesicles (SMVs), double membrane vesicles (DMVs) and multi membrane vesicles (MMVs) (*Doerflinger et al., 2017*) . The process by which the non-structural proteins induce the formation of these structures is unknown, but the NS1/2 and NS4 proteins are thought to be key to the process. We have previously reported that the NS1/2 protein from MNV and HuNoV interacts with the VapA and VapB proteins and that this interaction is required for efficient viral replication (*McCune et al., 2017*). The norovirus NS4 protein associates with lipid droplets and is able to drive the formation of SMVs and DMVs (*Doerflinger et al., 2017*), suggesting that it may regulate the biosynthetic pathways involved in lipid synthesis or intracellular vesicular transport pathways. Our characterisation of the replication complex indicates an intimate link with numerous proteins involved in vesicle transport and lipid metabolism (*Figure 2—figure supplement 1*). Given that NS1/2 and NS4 interact to form a complex (*Emmott et al., 2019*), it is likely they work in concert to drive the formation of membrane-bound replication complex by regulating the functions of multiple components, the characterisation of which will require further analysis.

In conclusion, our data adds significantly to the growing body of literature on the role of G3BP proteins in the life cycle of viruses and further extends the functional roles of G3BP1 to include the promotion of viral translation processes. We identify G3BP1 as a host protein that has a critical role in the life cycle of murine and human noroviruses, identifying the first cellular pro-viral protein with pan-norovirus activity. Furthermore, given the apparent importance of G3BP1 to an early stage of

the norovirus life cycle, this work suggests that targeting G3BP1 may hold future therapeutic potential.

## Grants

CBW was supported by NIH K08 AI128043 and Burroughs Wellcome Fund. MAM was supported by NIH U19 AI109725. IG is a Wellcome Senior Fellow. This work was supported by funding from the Wellcome Trust (Refs: 207498/Z/17/Z and 104914/Z/14/Z) and the Biotechnology and Biological Sciences Research Council UK (Refs: BB/N001176/1and BB/000943N/1). JBE was supported by a Churchill Scholarship.

# Materials and methods

## Cells

The murine microglial BV2 cell line (*Blasi et al., 1990*) was provided by Jennifer Pocock (University College London). BV2 cells were maintained in DMEM supplemented with 10% FCS (Biosera), 2 mM L-glutamine, 0.075% sodium bicarbonate (Gibco) and the antibiotics penicillin and streptomycin. BHK cells engineered to express T7 RNA polymerase (BSR-T7 cells, obtained from Karl-Klaus Conzelmann, Ludwid Maximillians University, Munich, Germany) were maintained in DMEM containing 10% FCS, penicillin (100 SI units/ml) and streptomycin (100 µg/ml), and 0.5 mg/ml G418. U2OS cells and derivatives of them were obtained from Nancy Kedersha (Harvard Medical School). All cell lines were screened for mycoplasma and confirmed as negative. The identity of the cell lines was not confirmed by STR profiling.

## Generation of G3BP1 KO cells

BV2 cells were cultured in DMEM containing 10% FBS and 1% HEPES. G3BP1 knockout BV2 cells were generated using two approaches. The clone 1B2 was generated by transiently transfected with Cas9 and a sgRNA (5TTCCCCGGCCCCGGCTGATGNGG) targeting exon 7 of G3BP1. BV2 cells were then single cell cloned and G3BP1 was sequenced by Illumina HiSeq. BV2 cells are polyploid at the G3BP1 locus as described previously (*Orchard et al., 2016*). Clone 1B2 also had three independent deletions at the sgRNA binding site resulting in deletions of 1, 2, and five base pairs respectively. The mutations introduced into the IB2 BV2 cell clone resulted in frame shifts at nucleotide positions 253, 254 and 244 and the absence of detectable G3BP1 protein as measured by western blot. G3BP1 knockout BV2 cell clones A, C and F were generated using an independent approach that relied on first generating a pool of three lentiviruses carrying guide RNAs TGTGCAACATG TCCGGGGCC, CAAACTCCCGCCCGACCAGC and TAGTCCCCTGCTGGTCGGGC targeting the first 100 bp of the coding sequence, cloned into pLentiCRISPRv2 (*Sanjana et al., 2014*). BV2 cells were then transduced with the pool of 3 lentiviruses, selected by puromycin treatment for 72 hr, prior to cloning by limiting dilution. Individual clones were then screened by western blot for the absence of G3BP1.

## G3BP1 complementation

Mouse G3BP1 cDNAs were subcloned into pCDH-MCS-T2A-puro-MSCV lentiviral vector (System Biosciences) by NEBuilder HiFi DNA assembly (New England Biolabs). Mouse G3BP1 was subcloned from pCM6-G3BP1 (MR207441; Origene). Mouse G3BP1 lentiviral constructs deficient in the C-terminal RGG domain (mG3BP1ΔRGG) and the RGG and RRM domains (mG3BP1ΔRGGRRM) were generated by Gibson cloning from the pCMV6-G3BP1 vector. Lentivirus was generated by co-transfecting pCDH-G3BP1-T2A-puro-MSCV with pCMV-VSV-G and pSPAX2 into 293 T cells with Trans-IT LT1 (Mirus Biosciences) per manufacture instructions. Two days post-transfection, supernatants were harvested, filtered through a 0.22 micron filter, and stored at −80C. Lentivirus encoding G3BP1 or an empty control was then used to transduce G3BP1 KO 1B2 BV2 cells. Two days post-transduction BV2 cells were selected with puromycin (2.5 ug/ml) for six days.

## MNV growth curves

To determine the effects of G3BP1 disruption on MNV replication G3BP1 WT, KO, or complemented cells were plated in each well of a flat bottom 96-well plate and then infected with either MNV

strains CW1, CW3, or CR6 as described in the text. Infected cells were flash frozen at −80°C at the times post infection indicated in the text. Viral replication was then assessed by plaque assay or TCID50 in BV2 cells as described in the text. In cases where the appearance of virus-induced cytopathic effect was examined, infected monolayers were either visualised by light microscopy directly or fixed with crystal violet in formalin, prior to washing and imaging.

## CRISPR screens

The CRISPR screen was performed similarly to that described previously (*Orchard et al., 2016*) with a number of modifications that included the use of the Brie gRNA library to reduce off target effects (*Doench et al., 2016*) and shorter infection times to improve the recovery of gRNAs that may also compromise cell viability. BV2 cells stably expressing Cas9 nuclease (*Orchard et al., 2016*) were transduced with the Brie library using previously described protocols (*Doench et al., 2016*). MNV strains CW3 and CR6 were used to infect BV2 CRISPR library at MOI five pfu/cell and cells were isolated 24 hr post infection and preparation of gDNA for sequencing as described previously (*Orchard et al., 2016*). The screen relies on the premise that guide RNAs targeting genes that are overrepresented following infection represent genes that when disrupted are protected against infection and therefore likely represent factors with pro-viral activity. Likewise, genes for which guide RNA are underrepresented suggest that infection had proceeded faster and the gene is anti-viral. Following sequencing, the data was analyzed by STARS method as previously described (*Doench et al., 2016*; *Orchard et al., 2016*). Visualisation of candidate genes was accomplished using R (RStudio, Inc, Boston, MA).

## Maintenance of SILAC cell lines

Stable isotope labelling of amino acids in cell culture of BV2 cells (SILAC, *Ong et al., 2002*), was carried out in high-glucose DMEM lacking arginine and lysine (Sigma-Aldrich), supplemented with dialyzed fetal bovine serum, 1% L-glutamine, 1X nonessential amino acids, 10 mM HEPES, and 1X penicillin/streptomycin. SILAC media were supplemented with Light (R0K0), Medium (R6K4) or Heavy (R10K8) Arginine and Lysine (Cambridge Isotope Laboratories). BV2 cells were maintained in SILAC medium for 2 weeks to ensure complete metabolic labelling of proteins. Labelling of HEK-293T cells was performed essentially as described for BV2 cells, with the omission of 10 mM HEPES and 1X non-essential amino acids from the cell culture media.

## DNA based recovery of murine norovirus

Experiments were performed according to previously published protocols (*Chaudhry et al., 2007*). Briefly, BSRT7 cells were infected to an MOI of 0.5–1 PFU/cell with fowlpox virus expressing T7 RNA polymerase. Cells were then transfected with a plasmid encoding the MNV full length clone, or a derivative thereof (e.g. pT7 MNV 383FLAG 3'Rz or pT7 MNV 2600FLAG 3'Rz, our FLAG-tagged virus constructs containing FLAG tags in either NS1/2 or NS4 respectively). MNV was harvested by freeze-thaw at 24 hr post-transfection.

To generate higher titre stocks, WT MNV, NS1/2-FLAG MNV, and NS4-FLAG MNV (*Thorne et al., 2012*) generated using the DNA based recovery method described above were passaged once in BV2 cells. After 80–90% of cells displayed visible cytopathic effects (CPE) of viral infection, flasks containing infected cells were frozen at −80°C. Flasks were frozen and thawed twice before cell debris was removed by centrifugation at 4000 rpm for 10 min in a benchtop centrifuge. Viruses were pelleted by centrifuging over a 30% sucrose cushion at 76,755xg in a SW32ti rotor for 2 hr at 4°C. Virus pellets were resuspended overnight in PBS to achieve 100-fold concentration. Concentrated virus was then passed through a 23-gauge blunt needle 15 times, and clarified by centrifugation at maximum speed in a benchtop microcentrifuge for 10 min. Supernatant aliquoted, and titrated prior to use.

## Infection of SILAC labelled BV2 cells with FLAG-tagged viruses

SILAC-labelled BV2 cells were infected with WT MNV, NS1/2-FLAG or NS4-FLAG viruses at an MOI of 10 TCID50 cell. Infections were performed in triplicate, using different combinations of SILAC-labelled BV2 cells each time to control for any impact of the SILAC labelling. Infected cells were then plated in the appropriate SILAC media. At 10 hr post-infection, cells were harvested by scraping,

and pelleted at 500xg for 5 min. Cells were then washed three times with ice-cold PBS, and were lysed in (0.5% Nonidet-P40 substitute, 10 mM Tris-HCl pH 7.5, 150 mM NaCl, 0.5 mM EDTA, 2 mM MgCl). Benzonase nuclease (Sigma-Aldrich) was added to lysis buffer to a concentration of 5 µl/ml to prevent nonspecific interactions mediated by RNA or DNA.

## Transfection of SILAC labelled HEK-293T cells with GFP-tagged VPg

SILAC-labelled HEK-293T cells were transfected with pEGFP-C1 (control) or derivatives thereof containing either human or murine norovirus VPg protein as described in *Emmott and Goodfellow (2014)*. GFP fusions of both wild-type protein or mutant VPg containing a mutation to inhibit initiation factor binding (MNV: F123A, HuNoV: F137A) were used. Cells were transfected using Lipofectamine 2000 (Life Technologies) according to the manufacturers protocol, using antibiotic-free SILAC media in place of Opti-mem. The experiment was performed in triplicate and SILAC labels switched in one of the replicates.

## FLAG and GFP-TRAP immunoprecipitation

FLAG immunoprecipitations were performed following the manufacturer's protocol (FLAG M2 beads, Sigma Aldrich) as described (*Thorne et al., 2012*). In brief, protein concentration in lysates was normalised using BCA. Lysates were then diluted with 1 vol of wash buffer (10 mM Tris-Cl pH 7.5, 150 mM NaCl, 0.5 mM EDTA). Equal volumes of anti-FLAG affinity gel were dispensed into either WT infected cell lysates, or lysates of cells infected with NS1/2-FLAG or NS4-FLAG. Binding was carried out overnight at 4°C with rotation. After binding, beads were washed three times with wash buffer. All liquid was carefully removed from each tube, before boiling in SDS-PAGE loading buffer for 10 min. GFP-trap immunoprecipitation of GFP-tagged VPg was accomplished using GFP-trap beads (Chromotek) per the manufacturer's protocol, as described (*Emmott and Goodfellow, 2014*). RNase cocktail (Ambion) was also included in the lysis buffer at a concentration of 5 µl/ml to prevent non-specific interactions mediated by RNA. In all cases, light, medium, and heavy-labelled proteins eluted from the beads for each experimental replicate were pooled together in a ratio of 1:1:1 before submission for mass spectrometry analysis at the University of Bristol Proteomics Facility.

## Mass spectrometry analysis

Mass spectrometry analysis was performed at the University of Bristol Proteomics Facility. In brief, samples were run into precast SDS-PAGE gels for 5 min, the entire sample cut from the gel as a single band, and then subjected to in-gel tryptic digestion including reduction and alkylation using a ProGest automated digestion unit. The resulting peptides were fractionated using a Dionex Ultimate 3000 nanoHPLC system in line with an LTQ-Orbitrap Velos or Orbitrap Tribrid Fusion mass spectrometer.

## Interpretation of SILAC proteomics data

Raw data files were processed and quantified using Maxquant v1.5.5.1 or 1.6.0.16 (Tyanova *et. al.* 2016). The GFP-VPg experiments were searched against the Uniprot human database (70,550 entries, downloaded September 19[th] 2016) plus a custom FASTA file containing the wild-type and mutant VPg sequences. The raw data, search results and FASTA files can be found as part of PRIDE submission PXD007585 (Reviewer username: reviewer75984@ebi.ac.uk, Password: BH2pTctW). The FLAG-virus experiments were searched against the Uniprot mouse database (Swiss-prot only, 16,966 entries, downloaded May 19[th] 2018) plus a custom FASTA file containing the various Murine norovirus proteins. The raw data, search results and FASTA files can be found as part of PRIDE submission PXD011779 (Reviewer username: reviewer49419@ebi.ac.uk, Password: eLYwivNP). Data were searched with default Maxquant parameters including upto two missed tryptic cleavages, oxidation of methionine and N-terminal acetylation as variable modifications, and carbamidomethylation of cysteine as a fixed modification. The data were searched against a reverse database and PSM and Protein FDR were set to 0.01. The requantify option was not selected.

GFP-VPg data were analysed as described previously (*Emmott and Goodfellow, 2014*). FLAG-virus experiments were analysed by computing the pairwise ratios of samples infected with NS1/2-FLAG or NS4-FLAG relative to WT MNV-infected controls. Log$_2$ SILAC ratios for proteins identified

in at least 2/3 replicates were averaged, and ratios for NS1/2-FLAG:WT and NS4-FLAG:WT were plotted for comparison of host cell proteins by viral replication complex-associated proteins.

## Assessment of virus-induced cytopathic effect

BV2 WT, KO G3BP1 or respective G3BP1 complemented cells as described in the text, were seeded onto 96 well plates and infected with serial 10-fold dilutions (starting at MOI = 10 TCID50/cell) of MNV (CW1) or EMCV. At 48 hr post-infection, cells were fixed in ice-cold methanol and stained with toluidine blue prior to washing and imaging.

## Cap-Sepharose purification for eIF4F complex

Cell lysates were prepared from BHK parental cells or BHK containing GI Norwalk virus (BHK-NV) replicon cells in cap-Sepharose lysis buffer (100 mM KCl, 0.1 mM EDTA, 10% glycerol, 2 mM $MgCl_2$, 20 mM HEPES, pH 7.6 in KOH) with 1% TX-100, proteinase and phosphatase inhibitor cocktails (Calbiochem). Cytoplasmic extracts were centrifuged and RNase treated for 15 min at room temperature. At least 1000 μg of the cell lysates were incubated with Sepharose beads coupled to 7-methylguanosine ($m^7$GTP, Jena Biosciences). Input cell lysates were collected for western blot analysis while the remaining were incubated overnight with continuous rotation at 4°C. The eIF4F-enriched complex was precipitated and washed two times with cap-sepharose lysis buffer. Bound proteins were eluted in 2x reducing SDS-PAGE samples buffer and resolved by SDS-PAGE prior to western blot.

## Human Norwalk virus colony formation assay

Total RNAs extracts from BHK or BHK-NV replicon-containing cells (*Kitano et al., 2018*) were pretreated with and without proteinase K (10 μg/ml) in 10 mM Tris, pH 8.0, 1.0 mM EDTA, 0.1 M NaCl, and 0.5% SDS. Pretreated RNAs were immediately purified using GenElute RNA purification columns (Sigma). Serial 10-fold dilutions of mock or proteinase K-treated RNAs were transfected in BHK cells and 24 hr post transfection, cells were passaged and maintained in growth media containing 0.5 mg/ml G418. Colonies began to form after 5 d and were allowed to grow until 14 d. All plates were harvested at day 14 and well-formed colonies were fixed in 10% formaldehyde and stained with toluidine blue. A similar protocol was followed to assess colony formation in U2OS cells with the exception that selections were maintained for up to 12 days post transfection. Where indicated, cell aliquots from each time point were collected for qRT-PCR analysis to assess viral RNA synthesis over time.

## RNA co-immunoprecipitation

Coimmunoprecipitation of viral RNA with RpS3 and IgG, as irrelevant control, was performed using BV2 WT and BV2ΔG3BP1 clone IB2 cells inoculated with MNV1 at MOI of 50 TCID50 per cell in the presence of 400 μM 2-CMC. Cell lysates equilibrated in EE buffer (50 mM HEPES, pH 7.0, 150 mM NaCl, 0.5% NP-40, 10% glycerol, 2.5 mM EGTA, 5 mM EDTA, 1 mM DTT, 100 ug/mL Heparin, and HALT protease and phosphatase inhibitor cocktail) as previously described were pre-incubated with antibody against either RpS3 or IgG for 12 hr at 4°C with continuous rotation. Protein A/G Ultralink resin slurry (Thermo Fisher Scientific) was then added into cell lysates and antibody mixture and incubated further for 12 hr at 4°C. Resin were washed with EE buffer 3x and the complex bound to the resin were eluted in 0.1M glycine, pH 2. Elutions were subjected to RNA extraction and bound viral RNA were quantitated by RT-qPCR.

## Polyribosome fractionation analysis

BV2 WT and BV2 ΔG3BP1 cells were seeded at a density of $7.5 \times 10^6$ cells per T-75 flask, and then either mock infected or infected with MNV1 (CW1) at MOI 3 TCID50 per cell in the presence of 2-CMC (400 μM) for each set of infection. After 1 hr, the inoculum was then removed; the cells were washed and maintained in growth media containing 2-CMC accordingly until the cells were harvested at 4 hr and 9 h p.i. Prior to harvesting, cells were treated with cycloheximide (CHX) for 10 mins at 37°C (Sigma-Aldrich; 100 μg/ml) and were rinsed with 5 ml of ice-cold PBS supplemented with CHX 100 μg/ml. Polysome lysis buffer [20 mM Tris-HCl pH 7.5, 150 mM NaCl, 5 mM MgCl, 1 mM DTT, 1% Triton X-100, 100 μg/ml cycloheximide, 25 U/ml TURBO DNase (Life Technologies)]

was used to lyse the cells. Lysates were clarified by centrifugation for 20 min at 13,000 $g$ at 4°C. Aliquots of the lysates were collected for BSA assay and qPCR analysis against MNV1 RNA before proceeding with fractionation. Input lysates were normalised to total protein concentration and RT-qPCR was used to confirm the levels of viral RNA in samples were comparable. Lysates were subjected next to 10–50% sucrose gradient centrifugation for 90 mins SW41Ti rotor at 190,000 $x$ $g$ at 4°C. The gradients were fractionated at 0.5 ml/min and the levels of RNA in each sample measured using an in line-254 nm spectrophotometer connected to a chart recorder. RNAs were extracted from each fraction, converted to cDNA and immediately used for qPCR. The distribution of viral RNA across the gradient was then calculated as percentage (%) of the viral RNA seen in WT BV2 cells using the reference gene (GAPDH) to obtain normalised values across the gradient. Samples were performed in duplicates on the same qPCR plate, and the observations were robust across three independent experiments. Data were collected using a ViiA 7 Real-Time PCR System (Applied Biosystems).

## Transfection of VPg-linked MNV RNA into BV2 cells
VPg-linked RNA purified from MNV-1 virus particles was transfected in BV2 cells using NEON as previously described (*Yunus et al., 2010*). Total cell lysates were harvested at 3 and 9 hr post transfection with RIPA buffer. 10 µg total lysates were analysed by 4–12% SDS-PAGE (Invitrogen) and antibodies against MNV, VPg, G3BP1 and GAPDH were used for detection using LI-COR Odyssey CLx. Virus yield was determined by $TCID_{50}$. For strand-specific qPCR detection of MNV RNA, total cellular RNA was extracted using GeneElute Mammalian Total RNA Miniprep kit (Sigma). RT-qPCR was performed as described previously (*Vashist et al., 2012b*).

## Purification of MNV VPg-linked RNA
BV2 cells were infected at an MOI = 0.01 $TCID_{50}$ per cell and harvested after ~30 post infection. Cell debris was removed by low speed centrifugation for 10 min and supernatant loaded onto 5 ml of 30% sucrose solution in PBS. MNV particles were pelleted using a SW32Ti rotor at 25,000 RPM for 4 hr at 4 °C. Virus was then resuspended in PBS and total RNA extracted from soluble fraction. Where detailed, the authenticity of the viral RNA was examined by nuclease digestion. 500 ng of viral RNA or plasmid DNA was treated with DNase I (10U, Roche), XrnI +RppH (1U XrnI +5U RppH, both from NEB) or RNase cocktail (0.5U RNase A + 20U RNase T1, ThermoFisher) at 37 °C for 10 min. Then analysed on 1% agarose gel.

## Preparation of BV2 S10 cytoplasmic extracts
Preparation of BV2 S10 extracts was based on a previously published protocol (*Rakotondrafara and Hentze, 2011*; *Castelló et al., 2006*). BV2 cells were harvested, washed with PBS, and lysed with 1x packed volume of hypotonic buffer containing 10 mM HEPES pH7.6, 10 mM potassium acetate, 0.5 mM magnesium acetate, 5 mM DTT, 1x protease inhibitors cocktail (EDTA-free, Roche). Cells were lysed on ice for 45 min, then passed through 25G and 27G needles to achieve >95% lysis. Cell lysates were then centrifuged at 10,000 $x$ $g$ for 10 min at 4 °C twice and the supernatant collected. The total protein concentration was measured by Bradford assay and normalised to 20 mg/ml before freezing at −80 °C until use. For micrococcal nuclease treatment, S10 extracts were thawed on ice, 1 mM calcium chloride and 200 unit/ml final concentrations of micrococcal nuclease (NEB). Cell lysates were incubated at 25 °C for 15 min before adding 3 mM final concentration of EGTA was added.

## In vitro translation of BV2 S10 lysates
In vitro translation assays were set up based on a previous protocol (*Favre and Trepo, 2001*). Translation reactions were set up in 12.5 µl total volume containing 5 µl BV2 S10 lysate, 2.5 µl 5X translation buffer, 0.25 µl of 5 mg/ml creatine kinase, 1.25 µl RRL, 0.225 µl of 5 M potassium acetate, 0.25 mM of 100 mM magnesium acetate, 5.13 µCi $^{35}$S-labelled methionine (PerkinElmer) and 10–100 ng/µl RNA as detailed in the text. 5X translation buffer contains 350 mM HEPES, 75 mM creatine phosphate, 10 mM ATP, 3.75 mM GTP, 100 µM amino acid minus methionine, 3.75 mM spermidine and 0.375 mM S-adenosyl-methionine. For control experiments using RRL (Promega), the reactions were set up according to manufacturer's instructions using 0.5–1 ng/µl RNA. Reactions were incubated at 30 °C for 90 min before addition of 12.5 µl trans-stop buffer containing 10 mM EDTA and 0.1 mg/ml

RNase A and incubated at room temperature for 10 min, then 25 µl 5X loading buffer was added to the reaction and heated at 95 °C for 5 min. 10 µl lysates were resolved in 15% SDS-PAGE and exposed to a phosphorimager screen and visualised using a TyphoonFLA7000 machine. For non-radioactive translation, 1.25 µl of 1 mM methionine was used instead of $^{35}$S-labelled methionine, and the reactions were stopped with 100 µl 1x passive lysis buffer (Promega) and the luminescence read using a GloMax luminometer (Promega).

## Acknowledgements

The authors would like to thank Skip Virgin (Washington University in St. Louis) for intellectual input and the provision of reagents and resources. Kate Heesom of the University of Bristol Proteomics facility for support with the mass spectrometry analysis and Nerea Irigoyen (University of Cambridge) for help with polysome profiling.

## Additional information

### Funding

| Funder | Grant reference number | Author |
|---|---|---|
| Wellcome | 207498/Z/17/Z | Myra Hosmillo<br>Jia Lu<br>James B Eaglesham<br>Xinjie Wang<br>Edward Emmott<br>Patricia Domingues<br>Yasmin Chaudhry<br>Tim J Fitzmaurice<br>Matthew KH Tung<br>Ian G Goodfellow |
| National Institutes of Health | AI128043 | Craig B Wilen |
| Biotechnology and Biological Sciences Research Council | BB/N001176/1 | Jia Lu<br>Ian G Goodfellow |
| Wellcome | 104914/Z/14/Z | Ian G Goodfellow |
| Burroughs Wellcome Fund | | Craig B Wilen |
| Churchill College, University of Cambridge | | James B Eaglesham |
| Biotechnology and Biological Sciences Research Council | BB/000943N/1 | Nicolas Locker |

The funders had no role in study design, data collection and interpretation, or the decision to submit the work for publication.

### Author contributions

Myra Hosmillo, Jia Lu, Formal analysis, Supervision, Validation, Investigation, Visualization, Methodology, Writing—original draft, Writing—review and editing; Michael R McAllaster, Conceptualization, Formal analysis, Validation, Investigation, Visualization, Methodology, Writing—original draft, Writing—review and editing; James B Eaglesham, Data curation, Formal analysis, Funding acquisition, Validation, Investigation, Visualization, Methodology, Writing—original draft, Writing—review and editing; Xinjie Wang, Validation, Investigation, Visualization, Methodology, Writing—review and editing; Edward Emmott, Conceptualization, Data curation, Formal analysis, Supervision, Validation, Investigation, Visualization, Methodology, Writing—review and editing; Patricia Domingues, Formal analysis, Investigation, Methodology, Writing—review and editing; Yasmin Chaudhry, Formal analysis, Investigation, Methodology; Tim J Fitzmaurice, Validation, Investigation, Methodology; Matthew KH Tung, Formal analysis, Investigation, Visualization, Methodology, Writing—review and editing; Marc Dominik Panas, Resources, Investigation, Methodology, Writing—review and editing; Gerald McInerney, Nicolas Locker, Conceptualization, Resources, Formal analysis, Supervision, Funding acquisition, Writing—review and editing; Craig B

Wilen, Conceptualization, Data curation, Formal analysis, Supervision, Funding acquisition, Validation, Investigation, Visualization, Methodology, Writing—original draft, Project administration, Writing—review and editing; Ian G Goodfellow, Conceptualization, Resources, Formal analysis, Supervision, Funding acquisition, Validation, Investigation, Visualization, Methodology, Writing—original draft, Project administration, Writing—review and editing

**Author ORCIDs**
Myra Hosmillo (iD) https://orcid.org/0000-0002-3514-7681
Tim J Fitzmaurice (iD) http://orcid.org/0000-0003-1403-2495
Marc Dominik Panas (iD) http://orcid.org/0000-0002-7373-0341
Ian G Goodfellow (iD) https://orcid.org/0000-0002-9483-510X

**Decision letter and Author response**
Decision letter https://doi.org/10.7554/eLife.46681.030
Author response https://doi.org/10.7554/eLife.46681.031

## Additional files

**Supplementary files**

• Supplementary file 1. Raw data associated with *Figure 1*. (**A**) Log2 SILAC ratio of proteins identified in Norwalk virus (NV) VPg GFP Trap pull downs. The experimental details are provided in the Materials and methods. SILAC ratios were computed by comparing the ratio of the peptides for each protein in the GFP control pull down to that obtained with the NV-GFP fusion protein. As described in the text each experiment was performed in three independent conditions varying the label in each biological sample. M/L – medium vs light sample, H/L – heavy vs light labelled sample and M/L – medium vs light sample. (**B**) As described in panel A but as obtained using the MNV VPg-GFP fusion protein as the bait.
DOI: https://doi.org/10.7554/eLife.46681.019

• Supplementary file 2. Raw data associated with *Figure 2*. (**A**) Log2 SILAC ratio of proteins identified anti-FLAG immunoprecipitations from cells infected with the NS1/2 tagged MNV. The experimental details are provided in the Materials and methods. SILAC ratios were computed by comparing the ratio of the peptides for each protein in the anti-Flag immunoprecipitations performed on cells infected with wild type MNV. As described in the text each experiment was performed in three independent conditions varying the label in each biological sample. M/L – medium vs light sample, H/L – heavy vs light labelled sample and M/L – medium vs light sample. The Log2 transformed values and the average values are shown. (**B**) As described in panel A but as obtained using the NS4-FLAG tagged MNV. C) Combined dataset obtained using NS1/2 and NS4 tagged viruses showing the average Log2 SILAC ratios.
DOI: https://doi.org/10.7554/eLife.46681.020

• Supplementary file 3. Gene ontology and cross comparison analysis of the data obtained in *Figure 2*. (**A**) Gene ontology of the host proteins found to be enriched by both NS1/2 and NS4. (**B**) PANTHER overexpression analysis of host proteins found to be enriched by both NS1/2 and NS4. (**C**) Raw data used for PANTHER analysis shown in panel B. (**D**) Colour code for data shown in panel C. (**E**) List of host proteins identified in previous studies as having (potential) roles in the norovirus life cycle along with their degree of overlap with the host proteins identified using NS1/2 and NS4. (**F**) Gene ontology analysis of host proteins identified as binding to the MNV NS1/2 protein and enriched using GFP-tagged MNV VPg.
DOI: https://doi.org/10.7554/eLife.46681.021

• Supplementary file 4. Raw data and further analysis of the data obtained in *Figure 3*. (**A**) List of genes ranked with a positive STARS obtained with the Brie CRISPR screen against MNV CW3. (**B**) List of genes ranked with a negative STARS obtained with the Brie CRISPR screen against MNV CW3. (**C**) As in panel A except using the MNV strain CR6. (**D**) As in panel B except using the MNV strain CR6. (**E**) Combined STARS ranking for the genes in both MNV CW3 and CR6 data sets. (**F**) Comparison of genes with positive and negative STARS values in the data sets obtained using MNV CW3 and CR6. (**G**) Comparison of data from panel A and C with the previous MNV CRISPR screens.

(**H**) Gene ontology overexpression analysis of genes identified in this study as having positive STARS values for both MNV CW3 and CR6.
DOI: https://doi.org/10.7554/eLife.46681.022

• Supplementary file 5. Comparison of data obtained from three screens to identify host factors involved in the norovirus life cycle. (**A**) Comparison of the data obtained using MNV VPg-GFP trap with the MNV NS1/2 and NS4-FLAG tagged purifications. (**B**) Comparison of the data obtained using the CRISPR screen and the MNV NS1/2 and NS4-FLAG tagged purifications. (**C**) Comparison of the data obtained from all three screens.
DOI: https://doi.org/10.7554/eLife.46681.023

• Transparent reporting form
DOI: https://doi.org/10.7554/eLife.46681.024

### Data availability

VPg proteomics raw data, search results and FASTA files can be found as part of PRIDE submission PXD007585. Flag-virus proteomics raw data, search results and FASTA files can be found as part of PRIDE submission PXD011779.

The following datasets were generated:

| Author(s) | Year | Dataset title | Dataset URL | Database and Identifier |
|---|---|---|---|---|
| Hosmillo M, Lu J, McAllaster MR, Eaglesham JB, Wang X, Emmott E, Domingues P, Chaudhry Y, Fitzmaurice TJ, Tung MKH, Panas M, McInerney G, Locker N, Wilen CB, Ian Goodfellow I | 2019 | VPg proteomics raw data | https://www.ebi.ac.uk/pride/archive/projects/PXD007585 | PRIDE PRoteomics IDEntifications, PXD007585 |
| Hosmillo M, Lu J, McAllaster MR, Eaglesham JB, Wang X, Emmott E, Domingues P, Chaudhry Y, Fitzmaurice TJ, Tung MKH, Panas M, McInerney G, Locker N, Wilen CB, Ian Goodfellow I | 2019 | Flag-virus proteomics raw data | https://www.ebi.ac.uk/pride/archive/projects/PXD011779 | PRIDE PRoteomics IDEntifications, PXD011779 |

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
