## [Decision Letter]

Thank you for submitting the interesting article, "Noroviruses subvert the core stress granule component G3BP1 to promote viral VPg-dependent translation", for consideration by *eLife*. Your article has been reviewed by three peer reviewers, including Karla Kirkegaard as the Reviewing and Senior Editor. The following individual involved in review of the submission has agreed to reveal his identity: Volker Thiel (Reviewer #1).

The reviewers have discussed the reviews with one another and the Reviewing Editor has drafted this decision to help you prepare a revised submission.

This manuscript does an excellent job of documenting three different methods to identify potential host factors in the growth of murine and human noroviruses. One of them focuses on proteins present in complexes that contain VPg, a protein that is covalently linked to the genome. Known to be a primer of viral RNA synthesis, an hypothesis that focuses this host factor search is that it is likely that VPg is also involved in translation of the viral genome. Of the many interesting candidates, they choose to study further a member of the stress granule complex, GCBP1. This is a good choice because of the abundant literature on both proviral and antiviral effects of this complex, so mechanistic detail for the observed proviral role is of great interest.

Overall, this is a data-rich study leading to a novel and very interesting function of G3BP1 during norovirus replication. Given the technical limitations associated with experimental systems for norovirus infection, this study will be a valuable resource for future research. Each screen contains a wealth of interesting hits that will stimulate the field of RNA virus research, and the authors are requested to include additional information on the individual screens.

While the authors present strong evidence for the critical role that G3BP1 plays in norovirus infection generally, the mechanistic studies showing that it functions to recruit 40S ribosomes to the initiation complex and that it provides a competitive advantage to viral genomes compared to cellular RNAs need strengthening. Specific comments below include the inclusion of additional data, more straightforward interpretation of the existing data, and increased clarity of presentation.

Major comments: Experimental

1) In Figure 9, the amount of RpS6 that is co-immunoprecipitated with the VpG fusion is so low that it is difficult to be completely convinced that it is higher in the presence than in the absence of G3BP1. Considering this is the key finding of the figure, the authors are encouraged to strengthen this finding, perhaps by examining other 40S subunit proteins. Additionally, a critical test that G3BP1 is required for VPg association with translational initiation complexes would be to repeat much of the data in Figure 1 in the absence of G3BP1.

2) Figure 11 as it currently stands cannot be evaluated because insufficient experimental details are provided. The authors should explain how experiments to analyze cap- and IRES-dependent translation were performed.

3) The interpretation that addition of cellular RNAs enhances the reduction of viral translation in the absence of G3BP1 is based on results in Figure 11D-E. These data are highly normalized and it is not convincing that the 80% reduction in viral translation is significantly different from 40-50% reduction in the absence of cellular RNAs. This would seem to require a large number of replicates and statistical stringency to conclude. Besides, this argument only indirectly supports an hypothesis that cellular and viral RNAs compete for translational machinery, which is only one explanation for the proposed preferential translation of viral RNAs. Thus, it could be deleted and replaced with an experiment that speaks more to translational efficiency, such as ribosomal occupancy or quantitation of protein synthesis in a pulsed labeling experiment.

Major comments: Interpretation and explication

1) The term "translational bias" might confuse some readers because "bias" in the context of translation is usually used to discuss codon preference. Perhaps "preferential translation" of viral over cellular RNA would be more clear.

2) The evidence that bias or preferential translation of viral RNA over cellular RNA observed in wild-type-infected cells is caused by GCBP1 is not yet convincing. It is clearly shown that translation of viral RNA is inhibited in the absence of GCBP1 and translation of host RNA is not. However, preferential translation or bias in productively infected cells means that GCBP1 makes viral translation 'better' than host translation. How would this be measured? Perhaps ribosomal occupancy of host mRNA in the presence and absence of viral infection, in the presence and absence of GCBP1? As it stands, GCBP1 has been shown to specifically promote viral translation, but not bias.

3) The sixth paragraph of the Discussion is a list of observations from the literature that is not easy to follow. It doesn't really matter who did what experiment. The point is that the basis for any observed preferential translation of viral vs. host capped messages could have several origins. There are several pieces of data that bear on this point that could be presented logically as consistent with different hypotheses. This is interesting and should be more clearly explained.

4) Determination of the norovirus replication complex proteome: This section is of high interest and the authors are encouraged to extend the data analysis. For example, in the last paragraph of the subsection “Determination of the norovirus replication complex proteome”, there is a very brief comment concerning the degree of overlap between the data using VPg as a bait protein and the NS1/2 tagged or NS4 tagged viruses. Please elucidate a bit more on what that implies.

5) Similarly, the CRISPR data set is of high interest and the authors are encouraged to extend the data analysis. Specifically, a comparison to previously published CRISPR screens could be provided. Are there some common themes, pathways or cellular processes amongst the putative pro- and anti-viral factors?

6) In the subsection “G3BP1 is required for the association of VPg with 40S ribosomal subunits”, the authors state that the interaction between VPg and eIF4G occurs via a direct interaction between the highly conserved C-terminal region in VPg and the central HEAT domain of eIF4G (Leen et al., 2016) and does not require any additional cellular cofactors, at least in vitro. These experiments were performed in translation-competent extracts, and thus contain many cellular proteins that may very well be conserved. Therefore, it is not shown that the 'interaction is direct', if this means that GCBP1 and VPg binding would occur in isolation, as opposed to both being part of a stable complex that contains multiple proteins. This does not matter for the very interesting findings of the paper, but does for the biochemical mechanism of ribosome recruitment.

---

## [Author Response]

[…] While the authors present strong evidence for the critical role that G3BP1 plays in norovirus infection generally, the mechanistic studies showing that it functions to recruit 40S ribosomes to the initiation complex and that it provides a competitive advantage to viral genomes compared to cellular RNAs need strengthening. Specific comments below include the inclusion of additional data, more straightforward interpretation of the existing data, and increased clarity of presentation.

We thank the reviewers and the editor for their time and their very constructive reviews of our manuscript. We note that the study is “data rich” and that reading it will have taken a considerable amount of their time – we are grateful for this. We have taken on board the comments and as a result we have substantially revised the text and included additional experimental data where appropriate. In hindsight we feel that the message that G3BP1 provides a competitive advantage to viral translation is confusing and one that is not particularly well supported by the data we can obtain using the experimental approaches currently available to us – we have therefore toned down this narrative. Instead we emphasise the conclusion that G3BP1 plays a role in the novel paradigm of viral VPg-dependent translation initiation. The further dissection of the precise role for G3BP1 requires a fully in vitro reconstituted norovirus translation initiation system using purified translation initiation factors and ribosomal subunits. This is an approach we are working on, but which we feel is beyond the scope of the current study.

Major comments: Experimental1) In Figure 9, the amount of RpS6 that is co-immunoprecipitated with the VpG fusion is so low that it is difficult to be completely convinced that it is higher in the presence than in the absence of G3BP1. Considering this is the key finding of the figure, the authors are encouraged to strengthen this finding, perhaps by examining other 40S subunit proteins. Additionally, a critical test that G3BP1 is required for VPg association with translational initiation complexes would be to repeat much of the data in Figure 1 in the absence of G3BP1.

We accept the reviewers comments re the blots and we have invested the bulk of our time and effort in trying to address this particular issue. As part of the revision process we have purchased many additional antibodies to ribosomal proteins, many of which have failed to work in one or more of the assays we have attempted.As the reviewers are probably aware, we are somewhat restricted by the nature of the experimental systems we have available for noroviruses – in this case we are using overexpressed VPg in transfected cells, which importantly is not linked to the viral RNA genome. We find that the complex formed between overexpressed VPg and the ribosomal proteins to be somewhat unstable, most likely due to the fact that the interaction within the complex occurs by VPg binding first to eIF4G, which in turn binds eIF3 and then the ribosomal subunits. In the context of the viral infection, this complex is likely stabilised via secondary interactions between the initiation factors and the viral RNA which is covalently linked to VPg. We have previously published that eIF4A bindings to the sapovirus RNA genome directly and it is already well-established in the literature that eIF4G makes secondary RNA-protein contacts. Therefore maintaining the stability of this ternary complex throughout the multiple steps of the IP using VPg alone is challenging. We have amended the text according to make this clear.

To address the reviewers comment further, we have repeated the experiments multiple times and have now replaced the previous panel with another panel and supplementary figures that now include RpS6, RpS6 and RpL4. In addition we include data that demonstrates that the interaction is largely restored in the ΔG3BP1 cell line reconstituted with wild type G3BP1. We accept that the levels of the ribosomal proteins detected by western blot are still low, and further confounded by their abundance and rather ‘sticky’ nature, but we feel this has to be taken in the light of the fact that the VPg expressed in this context is not linked to viral RNA and therefore the secondary interactions with RNA that would stabilise this complex are not present. Therefore in addition, we have assessed the amount of viral RNA associated with ribosomal proteins (i.e. RpS3) using RNA-IP. Again, we consistently see that in the absence of G3BP1, the amount of viral RNA associated with ribosomal proteins is significantly reduced. The new data is included in the revised Figure 9 and an additional Figure 9—figure supplement 1.

2) Figure 11 as it currently stands cannot be evaluated because insufficient experimental details are provided. The authors should explain how experiments to analyze cap- and IRES-dependent translation were performed.

See comments above and below re the modifications to Figure 11– we have significantly modified the composition of this figure in light of the reviewers comments and have now included a clearer explanation of the method used to analyse the cap and IRES-dependent translation in the cytoplasmic extracts.

3) The interpretation that addition of cellular RNAs enhances the reduction of viral translation in the absence of G3BP1 is based on results in Figure 11D-E. These data are highly normalized and it is not convincing that the 80% reduction in viral translation is significantly different from 40-50% reduction in the absence of cellular RNAs. This would seem to require a large number of replicates and statistical stringency to conclude. Besides, this argument only indirectly supports an hypothesis that cellular and viral RNAs compete for translational machinery, which is only one explanation for the proposed preferential translation of viral RNAs. Thus, it could be deleted and replaced with an experiments that speaks more to translational efficiency, such as ribosomal occupancy or quantitation of protein synthesis in a pulsed labeling experiment.

We agree with the reviewers comments and thank them for highlighting the conceptual challenges this data presents. In light of their comments we have removed the component of the figure where we assess the impact of adding back cellular RNAs to nuclease treated extracts as we feel that this is a too artefactual. We believe that in order to be confident on the role of G3BP1 in the preferential translation of viral RNA we will need to establish a fully in vitro reconstituted translation system using purified ribosomal subunits and translation initiation factors, a significant undertaking in its own right. We aim to do this as part of a future study and have therefore revised Figure 11 to include in vitro analyses performed using nuclease treated extracts (panels D and E) and non-nuclease treated extracts (panels F and G). We find in both cases the translation efficiency is reduced in the absence of G3BP1. Whilst the reduction is modest (~50%), this likely reflect the in vitronature of the assay and the potent effect of this reduction on viral replication is likely amplified over the multiple rounds of replication. The revised figure still support our hypothesis that G3BP1 promotes norovirus VPg-dependent translation. The text has been updated to reflect these changes.

Major comments: Interpretation and explication1) The term "translational bias" might confuse some readers because "bias" in the context of translation is usually used to discuss codon preference. Perhaps "preferential translation" of viral over cellular RNA would be more clear.

As detailed above, upon reflection we feel that this narrative is somewhat confusing and we have therefore de-emphasised this particular hypothesis. Instead we focus purely on the observation that in the absence of G3BP1, norovirus a RNA is translated less efficiently and is recruited to ribosomal subunits (and polysomes) less efficiently. This observation represents a new function for G3BP1. We have changed the text accordingly.

2) The evidence that bias or preferential translation of viral RNA over cellular RNA observed in wild-type-infected cells is caused by GCBP1 is not yet convincing. It is clearly shown that translation of viral RNA is inhibited in the absence of GCBP1 and translation of host RNA is not. However, preferential translation or bias in productively infected cells means that GCBP1 makes viral translation 'better' than host translation. How would this be measured? Perhaps ribosomal occupancy of host mRNA in the presence and absence of viral infection, in the presence and absence of GCBP1? As it stands, GCBP1 has been shown to specifically promote viral translation, but not bias.

See response to comments above. In response to the reviewers comments we have revised Figure 11 and de-emphasised the hypothesis that G3BP1 provides a translational advantage to viral RNA, simplifying the message that G3BP1 plays a key role in the novel paradigm of viral VPg-dependent translation used by noroviruses.

3) The sixth paragraph of the Discussion is a list of observations from the literature that is not easy to follow. It doesn't really matter who did what experiment. The point is that the basis for any observed preferential translation of viral vs. host capped messages could have several origins. There are several pieces of data that bear on this point that could be presented logically as consistent with different hypotheses. This is interesting and should be more clearly explained.

We have amended the text to remove the emphasis on who did what experiments and when they were performed, to improve the overall clarity of this paragraph.

4) Determination of the norovirus replication complex proteome: This section is of high interest and the authors are encouraged to extend the data analysis. For example, in the last paragraph of the subsection “Determination of the norovirus replication complex proteome”, there is a very brief comment concerning the degree of overlap between the data using VPg as a bait protein and the NS1/2 tagged or NS4 tagged viruses. Please elucidate a bit more on what that implies.

We acknowledge the value of this data to those in the field and the rather limited nature of our initial analyses. We have amended the supplementary figure to include additional analysis, added an additional supplementary figure and added additional text to the Results and Discussion section.

5) Similarly, the CRISPR data set is of high interest and the authors are encouraged to extend the data analysis. Specifically, a comparison to previously published CRISPR screens could be provided. Are there some common themes, pathways or cellular processes amongst the putative pro- and anti-viral factors?

We agree that the data obtained from the CRISPR screen will contain a wealth of information that will be invaluable to the field and accept that our initial submission failed to provide a suitable analysis of this. We have taken on board the reviewers comments and tried to address this without being too verbose. We have compared our CRISPR data with that previously published and provide a list of genes enriched in both in a revised Supplementary file 4 – we find surprisingly little overlap, likely reflecting the experimental conditions under which the screens were performed i.e. the previous screens were performed under much more stringent conditions that were unlikely to identify proteins that play a role in the norovirus life cycle but are not essential. We have also added additional data analysis to the Results section.

6) In the subsection “G3BP1 is required for the association of VPg with 40S ribosomal subunits”, the authors state that the interaction between VPg and eIF4G occurs via a direct interaction between the highly conserved C-terminal region in VPg and the central HEAT domain of eIF4G (Leen et al., 2016) and does not require any additional cellular cofactors, at least in vitro. These experiments were performed in translation-competent extracts, and thus contain many cellular proteins that may very well be conserved. Therefore, it is not shown that the 'interaction is direct', if this means that GCBP1 and VPg binding would occur in isolation, as opposed to both being part of a stable complex that contains multiple proteins. This does not matter for the very interesting findings of the paper, but does for the biochemical mechanism of ribosome recruitment.

We are unsure as to the meaning of the reviewers comment but wish to clarify that the previous study was performed using recombinant proteins where we were able to demonstrate that the small C-terminal domain of VPg binds directly to the central domain of eIF4G without the aid of any additional factors. We have amended the text to ensure this point is clearer.